# Adipose MSCs Suppress MCF7 and MDA-MB-231 Breast Cancer Metastasis and EMT Pathways Leading to Dormancy via Exosomal-miRNAs Following Co-Culture Interaction

**DOI:** 10.3390/ph14010008

**Published:** 2020-12-24

**Authors:** Norlaily Mohd Ali, Swee Keong Yeap, Wan Yong Ho, Lily Boo, Huynh Ky, Dilan Amila Satharasinghe, Sheau Wei Tan, Soon Keng Cheong, Hsien Da Huang, Kuan Chun Lan, Men Yee Chiew, Han Kiat Ong

**Affiliations:** 1Faculty of Medicine and Health Sciences, Universiti Tunku Abdul Rahman, Cheras 43000, Malaysia; norlailyma@gmail.com (N.M.A.); boolily83@gmail.com (L.B.); cheongsk@utar.edu.my (S.K.C.); 2Marine Biotechnology, China-ASEAN College of Marine Sciences, Xiamen University Malaysia Campus, Jalan Sunsuria, Bandar Sunsuria, Sepang, Selangor 43900, Malaysia; skyeap@xmu.edu.my; 3Faculty of Medicine and Health Sciences, University of Nottingham (Malaysia Campus), Semenyih 43500, Malaysia; WanYong.Ho@nottingham.edu.my; 4Department of Agriculture Genetics and Breeding, College of Agriculture and Applied Biology, Cantho University, Can Tho 900100, Vietnam; huynhky@gmail.com; 5Department of Basic Veterinary Sciences, Faculty of Veterinary Medicine and Animal Science, University of Peradeniya, Peradeniya 20400, Sri Lanka; das@pdn.ac.lk; 6Institute of Bioscience, Universiti Putra Malaysia, Serdang 43400, Malaysia; tansheauwei@gmail.com; 7Cryocord Sdn Bhd, Persiaran Cyberpoint Selatan, Cyberjaya 63000, Malaysia; 8School of Life and Health Sciences, Warshel Institute for Computational Biology, The Chinese University of Hong Kong, Shenzhen 518172, China; huanghsienda@cuhk.edu.cn; 9Institute of Molecular Medicine and Bioengineering, National Chiao Tung University, Hsinchu 30010, Taiwan; r35085866@gmail.com; 10Department of Biological Science and Technology, National Chiao Tung University, Hsinchu 30010, Taiwan; chiewmene@gmail.com

**Keywords:** breast cancer cells, mesenchymal stem cells, microRNA, co-culture, exosomes, metastasis, recurrence, tumor microenvironment

## Abstract

Globally, breast cancer is the most frequently diagnosed cancer in women, and it remains a substantial clinical challenge due to cancer relapse. The presence of a subpopulation of dormant breast cancer cells that survived chemotherapy and metastasized to distant organs may contribute to relapse. Tumor microenvironment (TME) plays a significant role as a niche in inducing cancer cells into dormancy as well as involves in the reversible epithelial-to-mesenchymal transition (EMT) into aggressive phenotype responsible for cancer-related mortality in patients. Mesenchymal stem cells (MSCs) are known to migrate to TME and interact with cancer cells via secretion of exosome- containing biomolecules, microRNA. Understanding of interaction between MSCs and cancer cells via exosomal miRNAs is important in determining the therapeutic role of MSC in treating breast cancer cells and relapse. In this study, exosomes were harvested from a medium of indirect co-culture of MCF7-luminal and MDA-MB-231-basal breast cancer cells (BCCs) subtypes with adipose MSCs. The interaction resulted in different exosomal miRNAs profiles that modulate essential signaling pathways and cell cycle arrest into dormancy via inhibition of metastasis and epithelial-to-mesenchymal transition (EMT). Overall, breast cancer cells displayed a change towards a more dormant-epithelial phenotype associated with lower rates of metastasis and higher chemoresistance. The study highlights the crucial roles of adipose MSCs in inducing dormancy and identifying miRNAs-dormancy related markers that could be used to identify the metastatic pattern, predict relapses in cancer patients and to be potential candidate targets for new targeted therapy.

## 1. Introduction

Breast cancer is the fifth leading cause of cancer mortality worldwide, with the highest prevalence diagnosed (24.2%) and cancer mortality (15%) among women [1]. Metastatic or the spread of tumor cells throughout the body remains the underlying cause of death in the majority of breast cancer patients [2]. Patients presenting with distant metastasis are usually diagnosed with Stage IV disease and are unlikely treatable. Approximately 30% of women with breast cancer report recurrence with regional lymph node metastases despite early detection and advanced technology in cancer treatment due to the mechanism of resistance and tumor heterogeneity [3]. Breast cancer was generally acknowledged as being heterogeneous in terms of phenotypic, genetic and epigenetic makeup, which is categorized into few subtypes. The classification is based on receptor expression and proliferative activity, with MCF7 being a luminal A-type and MDA-MB-231 as basal B-triple-negative breast cancer (TNBC) [4]. The difference in the degree of tolerance to therapies among heterogeneous BCCs may render therapies in eliminating some subset of cancer cells such as dormant cells, contributing to late recurrence, particularly in highly metastatic cells [5].

In the event of metastasis, cells start to lose contact and adhesion facilitated by epithelial-mesenchymal transition (EMT) that leads to the increase of invasion and migration of cancer cells [6]. Many studies have identified the crucial role of EMT in metastasis of breast cancer [7,8]. Tumor cells can enter dormancy, a state of hibernation when cells are under an unfavorable microenvironment. Following this, the cells undergo metastasis latency, and once the microenvironmental conditions are favorable, they switch into proliferating cells and initiate the event of recurrence [9]. Therefore, the tumor microenvironment plays a crucial role in the maintenance of dormancy as well as the recurrence of cancer cells.

The tumor microenvironment is composed of cellular and non-cellular compartments. Besides tumor cells, it consists of the extracellular matrix, tumor vasculature, Mesenchymal stem cells (MSCs), immune cells and fibroblast. In the tumor niche, MSCs are actively recruited to the site of inflammation and injury, where they express growth factors that expedite the regeneration of tissues [10]. Several studies have shown that MSCs from various sources release multiple cytokines, growth factors and MicroRNAs (miRNAs) that could be used to treat cancer or as a source of pro-tumor factors [2,11,12,13]. Interestingly, the interaction between tumor and surrounding cells, particularly adipose MSC in the tumor microenvironment, has been the highlight of cancer research involving targeted therapy such as autologous fat grafting in breast reconstruction post-surgery, stem cell transplant in regenerative medicine and anticancer cell-based drug [10,14].

Collecting evidence suggests that dysregulation of miRNAs expression is associated with tumor formation, growth and metastasis either as promoter or suppressor [2,11,12]. In contrast, miRNAs involved in EMT regulation are known as EMT inducer or EMT suppressor [7]. Much interest has been generated in targeting circulating miRNAs, widely described as extracellular miRNAs secreted into the microenvironment via exosomes. Exosomes are small membrane vesicles that encapsulate molecules such as protein, lipids, transcription factors, RNAs and microRNAs and are secreted by numerous cell types into the surrounding. Exosomes have been shown to transfer cancer-promoter, suppressor miRNAs and other biological molecules through cell–cell interaction [11]. Hence, exosomal miRNAs have tremendous potential biomarkers for cancer diagnosis, monitoring recurrence and treatment response.

Over the past few years, several studies involving interaction between MSCs and BCCs have been carried out; however, due to the complexity of the cancer mechanism, most of these studies have failed to show a consensus on MSC’s effect on cancer progression. Moreover, studies involving miRNAs have focused mainly on its specific effects on corresponding target genes instead of looking globally at an extensive network. Therefore, sequencing has become very advantageous in analyzing miRNAs profiles, particularly to address a larger sample size. From clinical practice, there are currently no available markers that are able to predict the risk of late recurrence and determine which dormant population will eventually develop aggressive phenotype or remain dormant [15]. Thus, there is a need for fundamental research in identifying molecular markers that are associated with the transition of cells in or out of the dormant stage for each breast cancer subtype to determine the prognosis and the therapeutic possibilities.

This study hypothesized that circulating miRNAs secreted by adipose MSCs could affect breast cancer cells’ metastasis and dormancy, further promoting MET regulations. The interactions between miRNAs signaling molecules and MET regulators were investigated to explain why the invasive cancer properties of MDA-MB-231 cells continue to recur relative to MCF7. Using a Transwell system of indirect co-culture as a model to recapitulate the interaction between adipose MSC and breast cancer cells, the present work aims to discover their effect on metastasis events and explore the interplay of miRNAs as the main signaling molecules couriered by exosomes during the interaction. To improve effective treatment, a better understanding of the miRNAs-signaling mechanism that affects cancer progression and recurrence may be beneficial for the management of different subtypes of breast cancer.

## 2. Results

### 2.1. Adipose MSC of Co-Culture Promotes Cancer Dormancy and Chemoresistance

A non-contact co-culture model using the Transwell system was set up to recapitulate the interaction of stromal adipose MSC with BCCs in the microenvironment (Figure 1A). Transwell inserts made of polycarbonate membranes with 1.0 μm pore size were used to separate the two different cells, which allowed them to interact through secreted signaling molecules in an immediate environment. Adipose MSC displayed fibroblast-like morphology with a long-spindle shape at passage 5. Positive staining of Oil Red O and Alizarin Red shows that the cells were able to convert into lipid vacuoles adipocytes and calcium deposition osteocytes a week after differentiation, respectively. Immunophenotyping analysis shows the majority of cells are positive for CD90, CD44 and CD105 surface antigen expression with >85% prior to co-culture (Figure 1B). This verified stem cell marker expression in isolated adipose MSCs.

To study the dormancy effect of MSC co-culture on cancer progression, the proliferation of MDA and MCF7 was assessed using CCK-8 colorimetric assay. We speculated that the effect on cancer growth might be correlated with the distribution of the cell cycle. After 48 h of co-culture, significant inhibition of MDA and MCF7 cell proliferation was found in the presence of MSC compared to when being cultured alone (Figure 1C). Furthermore, the morphology of the MDA cells shifted from spindle-shaped fibroblast to cobblestone epithelial-shape with scattered colonies and fewer adherences. In the case of MCF7, the shape mostly remains the same, but there is an increase in the appearance of apoptotic and detached cells. In addition to that, the analysis of the cell cycle showed significant (*p* < 0.05) G0/G1 phase arrest in both cancer cells (Figure 1D). This was accompanied by a decline in cell growth (S phase). Meanwhile, both MDA and MCF7 co-culture cells demonstrated higher IC50 values of doxorubicin (DOXO), tamoxifen, cisplatin and 5HNQ compared to non-co-culture cells after 48 h of incubation (Figure 1E). When grown in 3D spheres, cancer stem cell (CSC)-enriched condition medium, co-culture MCF7 formed small (~50 µm in length), densely packed spheres and extremely low in number compared to co-culture MDA where they formed large (70–170 µm in length), loose clusters of cells and significantly higher spheres forming ability (Figure 1F). After dissociated into single cells, both 3D co-culture spheres were able to grow into 2D monolayer cells once they are subjected to a supportive niche (Appendix A).

Gene expression of multiple drug resistance (MDR)-ABC transporter genes, CSC genes and DNA repair genes were screened after co-culture treatment. Two of three MDR genes, ABCC2 and ABCG2, were consistently upregulated in both cells. The expression of CSC-associated surface markers, CD44 and ALDH1, was significantly downregulated only in MDA cells. Meanwhile, two of four genes that are involved in DNA repair and cell cycle, PARP1 and CCND2, were significantly dysregulated in both cells (Figure 1G). Overall, all genes that involve in the development of chemoresistance and regulation of cell survival were significantly dysregulated attributed to cell–cell interaction between adipose MSCs and BCCs.

### 2.2. Adipose MSC of Co-Culture Suppresses Breast Cancer Metastasis through MET Transformation

Transwell assay demonstrated that co-culture with MSCs resulted in a decrease of migratory and invasiveness of breast cancer cells (Figure 2A,B). As shown in Figure 2C, co-culture delayed the migratory distance of cancer cells in the wound gap created in the scratch assay as compared to the control cells for the time course over 24 h. In conjunction with that, we speculate that MSCs may alter the expression of epithelial (CD24) and mesenchymal (CD44) surface markers and genes associated with epithelial (E-cadherin, OLCN) and mesenchymal (SNAIL, ZEB2, vimentin and SMAD4) regulatory networks. To explore more on the involvement of EMT/MET regulatory network in suppressing cancer progression, epithelial and mesenchymal surface markers gene expressions that are required for initiation of metastasis were evaluated. The majority of MCF7 cells expressed epithelial surface marker, CD24 before and after co-culture, yet no significant increase on the level of the mesenchymal marker, CD44 (less than 8%) (Figure 2D). Plus, stable expression of epithelial genes E-cadherin and OCLN and a significant reduction in all mesenchymal genes were detected (Figure 2E). In the case of MDA cells, MSCs increased the expression of CD24 by 32%, increased E-cadherin and OLCN by three and seven-fold, respectively, and reduced SMAD4. Overall, MSCs maintain the epithelial identity and suppress the expression of mesenchymal genes in MCF7 cells. Meanwhile, MSCs enhanced the expression of epithelial in MDA cells while suppressing the mesenchymal markers. In MDA, MSCs lead the transition of EMT to MET state, whereas, in MCF7, MSCs maintain the cells in a slow proliferating state accordingly.

### 2.3. Intracellular Transfer of Exosomes and RNAs from Adipose MSC to Breast Cancer Cells Facilitate Dormancy Acquisition and Metastasis Inhibition

Earlier, we found that culturing BCCs with MSCs reduced proliferation and metastasis of cancer cells, suggesting that a factor secreted by MSCs was responsible for the acquisition of the dormant state of MCF7 and MDA cells. Thus, exosomes secreted during the co-culture interaction were further investigated. Exosomes collected from supernatants of MSC, MCF7, MDA-MB-231 and the respective co-cultures were successfully isolated using serial centrifugation followed by membrane affinity spin column. The presence of exosomes was verified through the detection of enzymatic activity of the exosome membrane protein, AChE. The enzyme assessment of exosomes indicated a slightly higher number of vesicles secreted by MSC and co-culture cancer cells compared to BCC alone (Figure 3A). To confirm the status structure of exosomes, electron microscopy imaging (TEM) was carried out (Figure 3B). Representative images show that isolated vesicles are primarily exosomes enriched with tetraspanin membrane protein markers CD81, CD9, CD63 and TSG101 shown by conjugated-gold particles in all exosome samples. Next, the direction of exosomes and RNAs transfer was identified through fluorescence dye staining, Vybrant Dio and SYTO64. Culturing cells with fluorescence-labeled exosomes in only one type of cell and detecting the presence of fluorescences in cells in another chamber allowed the confirmation of exosomes passing through the Transwell filter. Interestingly, both unstained cancer cells fluoresce green and red after 48 h co-culture with pre-stained exosomes-derived adipose MSC. Meanwhile, minimal fluorescence signals were detected in MSC cells when BCCs were pre-stained and co-culture for 48 h with MSCs (Figure 3C). The staining provides an indication that MSCs interact with BCCs in a bidirectional way, yet the majority of the exosome particles that carry RNA molecules were secreted by MSCs (act as donor cells) into the microenvironment and were taken up by breast cancer cells (act as recipient cells).

To confirm the presence of two-way cross-talk, we treated BCCs with exosomes derived from the culture medium of; (i) MSC-derived exosomes, (ii) MCF7 co-culture-derived exosomes and (iii) MDA co-culture-derived exosomes. The results were analyzed against control (no addition of exosome). Comparable to indirect Transwell co-culture, the addition of exosomes aliquot from 48 h of co-culture resulted in a transformation of breast cancer phenotypes; changes in morphology, reduction in proliferation (Figure 4A), migration (Figure 4B), invasion (Figure 4C) and wound healing (Figure 4D). Meanwhile, lower inhibitory levels were detected in BCCs treated with exosomes derived from MSCs alone relative to co-culture-derived exosomes. Overall, exosomes derived from co-culture show closer resemblance of biological properties in BCCs with indirect co-culture rather than exosomes derived from MSCs alone. This explains that adipose MSCs transfer consensus of signaling molecules to BCCs via exosomes into the immediate microenvironment depending on cell-to-cell interaction signals.

### 2.4. The Distribution of Small RNAs Differs between Cells and Exosomes

To discover the role of exosomes in mediating communication between MSC and breast cancer cells, it is crucial to investigate the RNAs content inside the vesicles that may play a role in cancer dormancy. Total RNA was isolated from exosomes released by MSC, breast cancer and co-culture cells. Results of bioanalyzer show that exosomes and cells consist of a different composition of the RNA population. Ribosomal RNA (rRNA) population (28S and 18S subunits) is highly enriched in cellular RNA, while, for the most part of exosomes, consists of short nucleotides. (Figure 5A). The population of miRNAs in both cells and exosomes was confirmed to be present at less than 40 nucleotides, with exosomes enriched in tRNAs at approximately 60 nucleotides. Prior to sequencing using NEB and Illumina platforms, a minimum cutoff RNA integrity number (RIN) for cells was set to be a minimum of seven, while for exosomes was between one and three due to the scarce amount of rRNA detected in exosomal RNA. Variation in RNA size distribution was detected between exosome and cell. Intense bands of shorter than 200 nucleotides were observed in exosomes indicating a small RNAs population, while longer length (1.9–4 kb) nucleotides were observed in cellular RNAs (Figure 5B). After demultiplexing and trimming, the quality was assessed using FastQC > 30. All reads from 30 samples (including replicates) were mapped to known small RNAs of the human genome and annotated to mature miRNA. A small RNA population mainly consists of 5S rRNA, transfer RNA (tRNA), micro RNA (miRNA), small nucleolar RNA (snoRNA), messenger RNA (mRNA), small nuclear RNA (snRNA) and yRNA (Figure 5C). The proportion of small RNA population is different between exosomal and cellular RNA, where miRNA and tRNA dominate exosomal RNA and miRNA and rRNA dominate cellular RNA. This presumably explains that miRNAs are one of the major components that are being transported between the cells. Nevertheless, the “unknown” or unannotated sequence fragments accounting for a quarter of mappable reads in all cells and exosome libraries (11–34%) were secreted into exosomes by the cells, indicating a significant number of novel small RNAs or miRNAs candidates that could potentially have regulatory functions.

### 2.5. Alteration of miRNAs Expression Profile in Cells and Exosomes Following Co-Culture

Even though exosomes are enriched in small RNA fragments, the relative miRNA composition is smaller in exosomes (8–20%) than in cells (38–57%) (Figure 5C). Based on the differentially expressed miRNA heat map and hierarchical clustering, three distinct patterns between parental cells, co-culture cells and exosomes were detected (Figure 6A). Among the cells, parental MCF7 and MDA have a clear separation from MSC, co-culture MCF7 and co-culture MDA cells. The fact that MSC is in the same cluster as the co-culture cancer cells shows that the donor (MSC) and recipient (co-culture cancer) cells share a similar expression of miRNAs. The similarity of miRNAs expression pattern is higher in both dormant BCCs subtypes after co-culture compared with non-dormant BCCs of the same origin (parental). This suggests that both cancer subtypes share similar miRNAs dormant markers that can be exploited for diagnosis and prevention of cancer relapse.

To demonstrate the role of exosomes as a carrier to transport miRNAs, the content should closely resemble those of the parent and recipient cells. In this case, hierarchical clusters of co-culture exosomes are closer to MSCs than parental cancer cells, which portrays that the content of co-culture exosomes is mainly derived from MSC cells rather than the latter. Interestingly, even though exosomes libraries are clustered within the same hierarchical group, there is a distinct miRNAs signature between exosomes derived from BCCs subtypes, MSC and co-culture BCCs subtypes.

The change in miRNA levels of exosomes and cells following co-culture is crucial to be explored as it may contribute to the overall effect of MSC on breast cancer cell proliferation and migration. To do so, differential expression in co-culture exosomes miRNA profiles were normalized against MSC exosomes and overlapped with co-culture cells normalized against non-co-culture cancer cells, which resulted in the changes of consensus miRNAs (Figure 6B). Based on the constructed Venn diagram of the top 50 miRNAs differentially expressed with fold change >2 or <−2, 14 miRNAs were upregulated, and 13 miRNAs were downregulated in MCF7, and 9 miRNAs were upregulated, and 2 miRNAs were downregulated in MDA, respectively (Table 1). Of the 38 miRNAs differentially expressed in both cells, subsequent overlapping resulted in only five miRNAs that are common to both cancer cells (miR-200a-5p, miR-941, miR-629-5p, miR-10b-5p and miR-486-5p), indicating that both sets of dysregulated miRNAs exclusively target different sets of genes (Table 2). These five miRNAs are the dormancy signatures secreted by adipose MSCs during bidirectional interaction and may contribute to the breast cancer recurrence of both subtypes. Taken together, the content and regulation level of miRNAs in released MSC-exosomes were altered after co-culture interaction with cancer cells for 48 h, which sequentially affects the miRNAs content of recipient cancer cells.

**Table 1 pharmaceuticals-14-00008-t001:** List of differentially expressed miRNAs in exosomes (co-culture normalized against MSCs) overlapped with cells (co-culture normalized against BCCs). From the list, there are five miRNAs that are mutually dysregulated in both breast cancer subtypes.

Up/Down-Regulated miRNA	Exo-MSC Fold Change >2	*p*-Value	Cell-MCF7 Fold Change >2	*p*-Value
hsa-miR-200a-5p ^1^	4.99	0.006	9.84	0.000
hsa-miR-203a-3p	3.01	0.179	8.24	0.000
hsa-miR-941 ^1^	3.10	0.066	7.62	0.005
hsa-miR-200a-3p	3.50	0.062	6.93	0.000
hsa-miR-629-5p ^1^	2.89	0.100	6.86	0.000
hsa-miR-200b-3p	4.43	0.002	6.81	0.000
hsa-miR-589-5p	2.00	0.441	6.17	0.002
hsa-miR-200c-3p	5.30	0.012	5.69	0.002
hsa-miR-3615	3.32	0.122	5.60	0.002
hsa-miR-7-5p	3.20	0.005	4.73	0.000
hsa-miR-185-5p	3.91	0.017	3.65	0.001
hsa-miR-1268b	2.22	0.315	3.57	0.058
hsa-miR-1268a	2.22	0.315	3.57	0.058
hsa-miR-375	3.76	0.045	3.41	0.018
hsa-miR-222-5p	−2.06	0.232	−5.18	0.001
hsa-miR-10a-5p	−2.30	0.028	−5.24	0.000
hsa-miR-221-5p	−3.61	0.026	−5.54	0.000
hsa-miR-143-3p	−2.58	0.032	−5.87	0.000
hsa-miR-199b-3p	−2.00	0.259	−6.19	0.000
hsa-miR-199a-3p	−2.00	0.259	−6.20	0.000
hsa-miR-199a-5p	−2.31	0.173	−7.06	0.000
hsa-miR-100-5p	−2.28	0.000	−7.19	0.000
hsa-miR-486-5p ^1^	−2.89	0.004	−7.46	0.000
hsa-miR-125b-1-3p	−2.00	0.000	−7.89	0.000
hsa-miR-155-5p	−13.68	0.000	−9.34	0.000
hsa-miR-10b-5p ^1^	−2.80	0.011	−10.22	0.000
hsa-miR-224-5p	−3.30	0.001	−23.77	0.000
**Up/down-regulated miRNA**	**Exo-MSC** **Fold change >2**	***p*-value**	**Cell-MDA-MB-231** **Fold change >2**	***p*-value**
hsa-miR-941 ^1^	3.58	0.114	12.89	0.000
hsa-miR-629-5p ^1^	4.82	0.006	9.61	0.000
hsa-miR-146a-5p	6.31	0.000	8.46	0.000
hsa-miR-1180-3p	4.13	0.128	8.39	0.001
hsa-miR-1246	4.15	0.008	7.64	0.000
hsa-miR-1290	4.74	0.027	7.46	0.002
hsa-miR-200a-5p ^1^	3.05	0.150	6.85	0.000
hsa-miR-1301-3p	3.07	0.308	6.04	0.034
hsa-miR-7704	6.22	0.000	6	0.002
hsa-miR-486-5p ^1^	−2.72	0.052	−6.97	0.000
hsa-miR-10b-5p ^1^	−3.93	0.004	−11.16	0.000

^1^ MicroRNAs that are mutually dysregulated in both breast cancer subtypes.

**Table 2 pharmaceuticals-14-00008-t002:** Top 10 of exclusive dysregulated miRNA-targeted genes in co-culture of BCC subtypes.

Target Gene in MCF7	Gene Description	Dysregulated miRNAs	O/E Ratio
ACVR2A	Activin A receptor, type IIA	10	0.916248
CBL	Cbl proto-oncogene, E3 ubiquitin protein ligase	8	0.52331
CAPRIN1 ^1^	Cell cycle-associated protein 1	7	0.562918
MAP3K7 ^1^	Mitogen-activated protein kinase kinase kinase 7	6	1.13509
CDC42	Cell division cycle 42	6	0.777232
PTEN ^1^	Phosphatase and tensin homolog	6	0.633646
SMAD2 ^1^	SMAD family member 2	6	0.400451
IGF1R	Insulin-like growth factor 1 receptor	6	0.392482
ERBB4	v-erb-b2 avian erythroblastic leukemia viral oncogene homolog 4	6	0.372997
NFAT5 ^1^	Nuclear factor of activated T-cells 5, tonicity-responsive	6	0.308160
Target gene in MDA-MB-231	Gene description	Dysregulated miRNAs	O/E ratio
CADM1	cell adhesion molecule 1	3	0.841896
NFAT5 ^1^	nuclear factor of activated T-cells 5, tonicity-responsive	3	0.378196
GAB1	GRB2-associated binding protein 1	2	0.977964
MAP3K7 ^1^	mitogen-activated protein kinase kinase kinase 7	2	0.92871
BRCA1	breast cancer 1, early onset	2	0.768397
PTEN ^1^	phosphatase and tensin homolog	2	0.518440
ARHGAP5	Rho GTPase activating protein 5	2	0.494601
CD28	CD28 molecule	2	0.428873
CAPRIN1 ^1^	cell cycle associated protein 1	2	0.394774
SMAD2 ^1^	SMAD family member 2	2	0.327642

^1^ Predicted genes highlighted in bold are mutually targeted by miRNAs in both cancer cells.

**Figure 6 pharmaceuticals-14-00008-f006:**
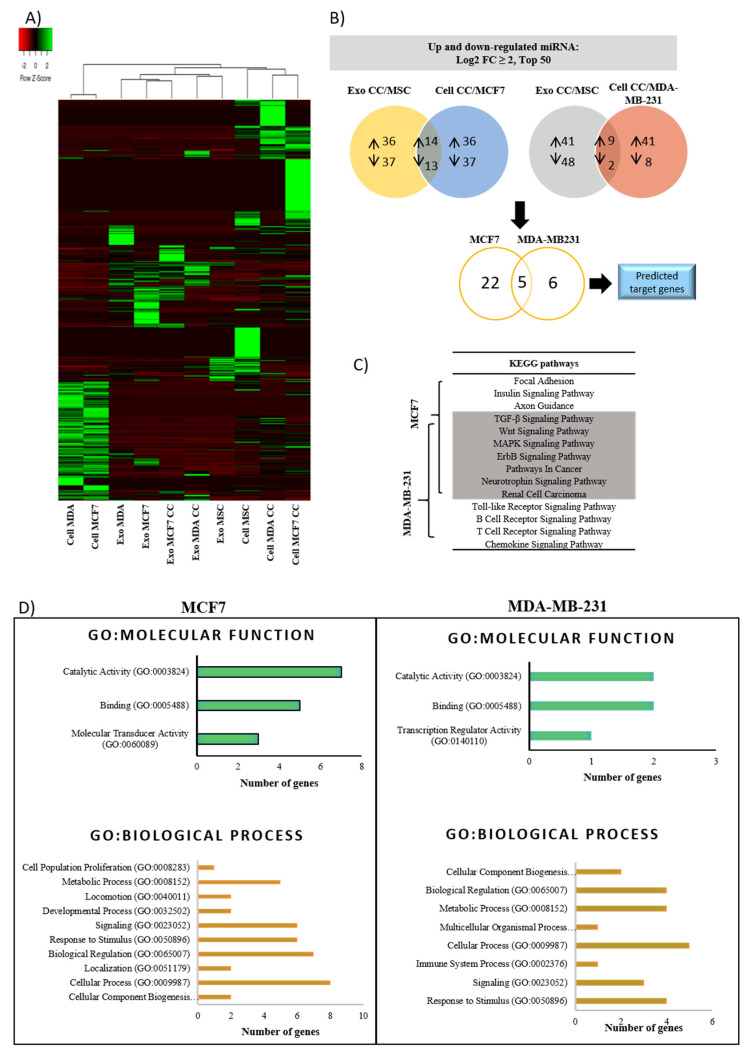
Differential miRNA expression and bioinformatic analysis of predicted target genes. (**A**) Heat maps and dendrograms generated by hierarchical clustering of miRNAs expression profiles signifies the different expression patterns and clusters of cells and exosomes. (**B**) The Venn diagram represents the differentially expressed miRNA (up- and downregulated) in exosomes (co-culture normalized against MSCs) overlapped with cells (co-culture normalized against BCCs). Further overlapping of the mutually dysregulated miRNA generates five miRNAs shared by both cancer cells (highlighted in Table 3). Predicted target genes were generated from the dysregulated miRNAs (Table 4). (**C**) Ten most enriched Kyoto Encyclopedia of Genes and Genomes (KEGG) pathways displaying mutual signaling pathways shared by both as well as exclusive pathways according to BCCs type. (**D**) Most significant gene ontology (GO) annotation is associated with molecular function and biological process, respectively, along with the number of genes that matched each GO term.

### 2.6. Biological Pathway Potentially Influenced by Dysregulated miRNAs-Mediated Breast Cancer Cells Dormancy

Earlier, it is noted that the majority of exosome trafficking directed from MSCs towards BCCs contributed substantial dysregulation in miRNAs expression of recipient cancer cells. Thus, we further analyzed the potential biological function of that direction for both MCF7 and MDA cells. MiRNAs regulate the expression of specific genes via hybridization to mRNAs to promote their degradation in order to inhibit their translation or both. Bioinformatics searches showed a total of 3857 putative target genes were identified in MCF7 and 832 in MDA, and 554 were shared by both. Among them, the top 10 targeted genes with highly significant O/E ratios are shown in Table 4. From that, five of the target genes were mutually predicted in both cells. Predicted genes targeted by dysregulated oncomiRs or tumor suppressor miRNAs were mainly involved in cell cycle regulators such as CAPRIN1, CDC42, PTEN, IGF1R, BRCA1 and CD28, thereby controlling cancer dormancy. Other predicted genes such as CBL, MAP3K7, SMAD2 and ERBB4 regulate multiple cellular processes; apoptosis and cell proliferation. Meanwhile, CADM1 and NFAT5 are critically involved in the migration and invasion of cancer cells.

Of the top 10 predicted targets in the top 10 GO terms in molecular function (MF), the percentage of binding and catalytic activity are the same for both cells (80%); however, 20% of the genes differed in terms of molecular transducer activity in MCF7 and transcription regulator activity in MDA cells (Figure 6C). The predicted genes associated with the biological process (BP) for both cell types were largely enriched in cellular process, biological regulation, metabolic process, response to stimulus and signaling.

Further, KEGG annotation revealed the exclusive and shared pathways of the ten most enriched KEGG pathways in both BCC types (Figure 6D). The shared pathways are related to tumor development and EMT related pathways such as TGF-β and Wnt signaling, cancer classical pathways (ErbB, MAPK, pathways in cancer) and pathways related to major intracellular signaling (neurotrophin signaling). MCF7 exclusively targeted pathways related to proliferation and EMT/MET, such as axon guidance and focal adhesion together with cancer metabolic processes (insulin signaling pathways). On the other hand, MDA targeted pathways enriched in physiological functions such as toll-like receptor (TLR), B cell receptor and T cell receptor and protein related to cancers (chemokines signaling pathways). Although some of the cascade and signaling pathways are dissimilar between two cells, they were all crucially involved in the regulation of cancer proliferation, metastasis and dormancy, leading to resistance. Taken together, exosomes from co-culture mediate the transfer of miRNAs that contribute to the dormant state of BCCs.

### 2.7. miR-941 Is Specifically Upregulated in Co-Culture Exosomes and BCCs

Among the commonly dysregulated miRNAs, miR-941 (greatest fold change in recipient cells) was selected as an upregulated miRNA marker that could represent both BCC types, which is significantly involved in exosomal trafficking. In addition, we selected miR-941 on the basis that it was previously reported to contribute to the suppression of cell proliferation, migration and invasion of hepatoma and gastric cancer cells [16,17]. Specific mir-941 miRNA-pathway network analyses showed that miR-941 participates in the regulation of B and T-cell receptor signaling, ErbB and MAPK signaling pathways (Appendix A). To confirm our selection of miR-941 and validate the findings of sequencing, the miRNAs expression level in cells and exosomes was performed using qRT-PCR. As compared to control, miR-941 was significantly upregulated (** *p* < 0.01, *** *p* < 0.001) in co-culture BCCs and exosomes, thereby is a potential upregulated miRNA diagnostic marker (Figure 7A).

As for sequencing validation, two mutual miRNAs expressed in both MCF7 and MDA cells, miR-941 and miR-10b-5p, were selected for validation. In addition, another three miRNAs (miR-760, miR-146a-5p and miR-205-5p) were chosen based on their suppression ability on breast cancer metastasis. An internal control RNU6b was employed for the assay. Overall, Figure 7B shows that miR-941 was the most steadily upregulated, whereas miR-10b-5p was the only downregulated miRNA of validated qPCR analysis. Meanwhile, miR-146a-5p was highly upregulated in the co-culture of MDA cells and exosomes but was downregulated in MCF7 cells. The results suggest that MSCs can selectively signal the secretion of specific miRNAs into the microenvironment via exosomes depending on the breast cancer subtype. Similarly, breast cancer cells can selectively uptake specific miRNAs depending on the breast cancer subtype. In conclusion, qRT-PCR results correlated well with sequencing data, indicating the reliability of sequencing-based expression analysis. MSCs secrete exosomes that suppress metastasis through miRNAs specific effect impeding migration, invasion and proliferation activities of breast cancer cells.

### 2.8. miR-941 Overexpression Suppress BCCs Viability and Metastasis via MET Regulations

We next sought to determine the role of miR-941 as tumor suppressor miRNA and a potential biomarker for BCCs survival. We overexpressed miR-941 mimic into two BCCs subtypes for 48 h and assessed their relative levels. As shown in Figure 8A, both cells have successfully been transfected and induced with significantly higher levels of miR-941 than the negative control. Overexpression of miR-941 has negatively impacted the viability and proliferation rate of both subtypes (Figure 8B). Furthermore, overexpression of miR-941 reduced cell migration (Figure 8C) and invasion (Figure 8D). In wound healing assay, wound closure by miR-941 cells transfected with the negative control was almost complete by 12/24 h and comparable to the untreated controls. On the contrary, both miR-941 mimic transfected cells migrated into the wound at a much slower rate and were significantly impaired in their ability to repopulate wounded areas when compared to untreated and negative control cells (Figure 8E). In general, overexpression of miR-941 developed a similar, low metastatic potential in both BCCs subtype and was comparable to co-culture cells. To identify the role of miR-941-induced MET regulation, epithelial and mesenchymal markers were screened. Overexpression of miR-941 significantly enhanced E-cadherin and impaired vimentin, SMAD4 and SNAI1 expression in both cells (Figure 8F). OCLN expression was enhanced in MDA cells, but not MCF7 cells. Meanwhile, the expression of the ZEB2 gene was suppressed only in transfected MCF7 cells. Taken together, these observations illustrate that miR-941 may facilitate the inhibition of breast cancer cell migration and invasion via MET regulation and strongly suggest that miR-941 has tumor suppressor function in both BCCs subtypes.

## 3. Discussion

In breast cancer research, MCF7 and MDA-MB-231 cells are the typical cell lines of two clinical breast tumor subtypes where their regulation depends on their interaction with surrounding cells that present in the tumor microenvironment. The interaction can affect cancer progression and thus influences patient prognosis and survival. Although adipose MSCs represent the most prominent cell type in the breast tumor microenvironment, very little attention was given to the adipose MSC population. Interestingly, growing research interest was noted in exploiting adipose MSCs in the cancer biology field due to their dual role as cancer promoters and suppressors [11,18]. Breast cancer cells are likely to disseminate during cancer progression and interact with MSCs in TME, releasing molecular signals contributing to the emergence of a subpopulation of cells that may go into a dormancy stage. These small subpopulations of cells are able to avoid chemotherapy treatment and reawaken in response to signals and transform into an aggressive phenotype, resulting in recurrence and metastasis. The risk of developing distant metastasis may vary over time across different subgroups of breast cancer cells.

Several types of in vitro co-culture models have been developed to study the interaction between cancer cells and microenvironment components such as extracellular matrix (ECM), fibroblast and MSCs, where these components have demonstrated the importance of the microenvironment in governing cancer cell growth [19,20,21]. The most common interaction models used to elucidate the interaction between two distinct cells include direct co-culture [19] and indirect co-culture using culture insert separating cells in space but sharing similar culture conditions, similar to our study [22]. There are also previous studies that have been carried out using conditioned mediums from MSCs and were incubated with cancer cells [23,24]. However, this method lacks the feedback-loop-dependent signaling or bi-directional cross-talk. There is also a 3D co-culture model established to recapitulate the complexity of interaction between BCCs and bone marrow MSCs, where it promotes dormancy [21,25]. Even though in vivo, the animal is a better model in representing the complexity and significance of the interaction between the cells, in vitro model is still required for preliminary study. Very few models have been developed concentrating on cellular dormancy and metastasis property [12,22]. Despite remarkable findings, the interaction effects of MSCs on BCC subtypes remain unanswered.

In this study, an indirect co-culture Transwell system was used to simulate and investigate non-contact interaction between adipose MSCs and BCC subtypes affecting cancer progression and metastasis. The findings demonstrated that when both BCCs interact with adipose MSCs indirectly, a reduction in cell proliferation, drastic change in cell morphology of MDA cells and induction of cell cycle arrest was noted. Tumor dormancy was reported to take place when cancer cells exit the cell cycle, survive in a quiescence state and cease to divide in an unfavorable microenvironment [26]. This implies that adipose MSCs promote dormancy of BCCs by suppressing cancer proliferation mediated by cell cycle arrest. The present findings were consistent with that reported by [26] where umbilical cord MSCs successfully prevented lung cancer cycle regulation, decreased cellular activity and increased the dormant population. The decrease in metabolic and cellular activities of dormant cells can be further linked to the increase in drug resistance against doxorubicin, tamoxifen, cisplatin and 5HNQ. This could be due to the design of drugs that target the rapidly proliferating cells, resulting in less effective chemotherapy drugs in eliminating resting dormant cells [9]. Moreover, it was found that the adipose MSCs served as the guardians for the BCCs, thus protecting them from chemotherapy effects. It was previously described that CD44+/CD24− breast cancer stem cells (CSCs) have a slow proliferating population, tumorigenicity and higher resistance to drugs [27]. However, the associated features of CSCs do not indicate dormancy in co-culture of BCCs with MSCs where the aldehyde dehydrogenase-1 (ALDH1), a functional regulator of CSCs and the surface abundance of the CD44+/CD24− cell population, a characteristic marker of breast CSCs significantly decreased in co-culture MDA cells that acquired dormant phenotypes.

Additionally, the proliferative activity of both co-culture subtypes (MCF7 and MDA-MB-231) was restored when the cells were placed in the original culture condition. The results supported the concept of dormancy, where cells are capable of reentering cell regulation when conditions become favorable [12]. When subjected to the co-cultured BCCs in a 3D sphere CSCs-supplemented model, MDA cells displayed enrichment of CSC compared to MCF7. The MCF7 cells maintained their slow-proliferation and epithelial characteristic in contrast to MDA cells that were able to exit dormancy and restore their mesenchymal appearances once removed from the inhibitory niche. It was also noted that co-cultured MDA cells were able to form spheres much more easily after reverting to their proliferating state. On the contrary, MCF7 had a lower sphere formation capability as compared to its parental counterpart after exiting the resting stage. This explains the higher recurrence and the lowest survival rate of MDA-basal subtype in the first 5 years of treatment in contrast to lower recurrence and higher survival period (>10 years) in MCF7-luminal patients [28]. This confirms that adipose MSCs serve as an inhibitory niche for breast cancer, which results in a temporary cell-cycle arrest. At the same time, the upregulation of drug resistance gene ABCG2, DNA repair gene PARP1 and regulator Cyclin D2 (CCND2) indicates the resistance to apoptosis leading to cell survival as supported by previous finding [26] which suggested the association between drug resistance and dormancy that represent an inhibitory phenotype of BCCs.

Mechanisms by which adipose MSCs inhibit BCCs proliferation through the transfer of exosomes carrying miRNAs from MSCs to BCCs were elucidated in this study. Changes in microenvironment following co-culture translate into rearrangements in the miRNA cargo within exosomes. MicroRNAs, which can be found in cells and exosomes, were shown to act as signaling molecules that control entire gene networks and were commonly expressed in a cell-type-specific and cancer subtype-specific manner [4]. With the use of fluorescence tracking dyes in the present study, it was discovered that throughout co-culture interaction, cells communicate via the transfer of exosomes directionally, mainly from MSCs to BCCs, creating a feedback loop. The direction of the transmission depends on the cell-specific interaction and the signal released by the cells [29]. Findings from this study further revealed that cell-to-cell interaction is the key to communication between MSCs and BCCs that inhibited cancer progression and facilitated dormancy acquisition. When BCCs were subjected to exosomes isolated from MSCs without co-culture, a less inhibitory effect was observed relative to co-culture-derived exosomes. The inhibition effect of co-cultured exosomes on proliferation, metastasis, invasion, and wound healing property closely resembles the inhibitory activity of exosomes from indirect Transwell culture compared to non-co-cultured exosomes. This event indicates that inhibition effects on BCCs are majorly attributed to the soluble factors released during the interaction between those two cell types in the same microenvironment. The soluble factors that are readily excreted by MSCs without interaction with cancer cells are also contributing to the inhibitory effect but to a lesser degree. This confirms that interaction with surrounding cells in TME is important for tumor progression and metastasis.

Interpretation of the miRNAs heat map profiles showed an interaction between MSCs and BCCs resulted in different clusters of miRNAs being exported, transported and received by recipient cells compared to miRNAs-derived exosomes from MSC alone, which may cause the disparity in cancer suppression level and dormancy effects among these cells (Figure 5A). Results from this study were consistent with that reported previously for a related study on the implication of miRNAs from MSCs-derived exosomes, which showed induction of cycling dormancy and early BCC quiescent in the bone marrow that gives rise to drug resistance. Nevertheless, the distinction in BCC subtypes was not emphasized [22].

The latest technique of transcriptomic study using the next generation sequencing (NGS) has shown the differences in miRNAs and gene expression profiles between the subtypes, which explained the stronger invasiveness of the MDA-MB-231 cell line compared to MCF-7 cells [4]. Using a similar platform, miRNAs expression of indirect co-culture exosomes and cells were profiled in this study to investigate the effects of miRNAs specific subtypes signaling mechanism on cancer proliferation and dormancy. While miRNAs were found to be one of the main RNA populations involved in a cell-to-cell transfer, sequencing results also revealed a distinct proportion of small RNA population in exosomes and cells. Furthermore, global microRNA expression profiles exhibited differential and subtype-specific microRNA expression profiles of dormant BCCs upon co-culture. Previous studies have highlighted the differences between MCF7 luminal and MDA-MB-231 TNBC in terms of their hormone receptors where TNBC are defined by the absence of estrogen, progesterone and HER-2 expressions, making them unresponsive to hormone and endocrine treatments and hard to treat. TNBC accounted for 10–20% of all breast carcinoma and prone to relapse compared to other subtypes. Their highly invasive and metastatic behavior, as well as the ability to stay dormant, making it easier to avoid chemotherapy and require special treatment approaches [4]. In the present study, we have identified miRNAs involves in inducing and maintaining cancer dormancy. Those miRNAs are targeting genes involves in cell cycle regulation and pathways associated with cell growth maintenance (Table 2). Seven miRNAs were found to regulate cell-cycle associated protein 1 (CAPRIN1) in MCF7 while two miRNAs were detected in MDA cells. This proves that more miRNAs are implicates in regulating the cell cycle and dormancy state of MCF7, making them stay longer in the dormancy state. A good strategy to lower the incidence of cancer relapse is by blocking the entry of BCCs into the dormant phase; thereby, cancer cells can remain in the proliferative state and would able to be eradicated using chemotherapy. Another way is by inhibiting the dormant cancer cells from entry into the cell cycle. Both ways can be achieved using miRNAs dormancy targeted therapy.

The fundamental step in cancer metastasis involves epithelial-to-mesenchymal transition (EMT), where MSCs have been reported to affect the transition in the invasive and metastatic type of cancers [30]. A mesenchymal phenotype is characterized by its increased motility, metastatic expansion and all the essential adaptations required in facilitating circulatory survival. Conversely, mesenchymal-to-epithelial (MET) is the reverse of EMT where cells acquire epithelial features and lose mesenchymal characteristics, including spindle-shaped and flattened phenotypes. The transition was known to impact cells in a metastatic niche to develop secondary tumors and has been associated with the acquisition of chemoresistance and recurrence [31]. Data from this study revealed that dysregulation of miRNAs subset in co-culture MCF7 cells did not only affect the growth of BCCs but also metastasis events by inhibiting EMT regulation. As part of MET, dysregulation of miRNA maintained the epithelial phenotype by blocking mesenchymal transcriptional regulators, specifically ZEB-2, vimentin, SMAD4 and SNAI1. The substantial miRNAs that have been dysregulated in co-culture MCF7 includes tumor suppressor miRNAs (miR-200 family, miR-941, miR-629-5p, miR-7, and miR-185-5p) and oncomiRs (miR-155-5p, miR-224-5p, miR-486-5p and miR-10b) (Table 3). Meanwhile, in co-culture MDA cells, tumor suppressor miRNAs (miR-146a-5p, miR-941, and miR-629-5p) and oncomiRs (miR-10b and miR-486-5p) were dysregulated. This resulted in the repression of vimentin and SMAD4 of mesenchymal transcription factor and substantial augmentation of OCLN and E-cadherin of epithelial transcriptional regulators. Increased expression of tumor suppressor miRNAs and reduced expression of oncogenic miRNAs in co-culture cells prevented cancer progression by miRNAs’ participation in a complex regulatory network, which affected the epithelial acquisition. An increase in epithelial gene expression and reduction in mesenchymal gene expression has led to the transition of mesenchymal to epithelial in MDA cells and maintenance of epithelial characteristics in MCF7 co-cultured cells.

Among the cluster of dysregulated miRNAs in co-culture MCF7 cells, the miR-200 family has been identified as the master key where it has been linked to chemoresistance, tumor metastasis and dormancy by targeting the signaling pathways including Wnt and transforming growth factor β (TGF-β), thus impeding metastasis, cell adhesion and epithelial-to-mesenchymal transition (EMT). As an EMT inhibitor, miR-200 targets ZEB2 transcription factors controlled the expression of gene clusters, including E-cadherin and vimentin [32]. Likewise, the downregulation of oncomiR-155 contributed to the increase in chemoresistance through TGF-β-induced MET and MAPK signaling pathways [33]. In addition, it has been reported that upregulation of miR-7 and miR-185 contributed to a similar outcome where miR-7 was negatively correlated with vimentin mRNA levels in breast carcinoma tissues by inhibiting EMT via focal adhesion pathway [4,34,35]. Meanwhile, in MDA, selective enrichment of miR-146a-5p was found in co-culture exosomes and cells, suggesting a possible involvement of miR-146a in pathogenesis and recurrence. The miR-146a-5p implicates toll-like receptors and chemokines signaling pathways that prompt the entry of cells into dormancy. The present result was in agreement with that reported in the previous study where miR-146a-mediated downregulation of SMAD4 target gene could inhibit proliferation, migration, invasion and EMT in TNBC via TGF-β and Notch pathways [36]. Another study has found that serum miR-146a level was correlated with drug resistance and higher recurrence, while their expression was classified as subtype-specific in breast cancer [37]. Therefore, a therapeutic approach targeting miR146a could be a promising treatment for high recurrence and poor prognosis in MDA cells. The output from cluster analysis of microRNA expression in BCCs subtypes in this study showed a complete separation of luminal and basal fractions in two distinct clusters.

Apart from the uniquely expressed miRNAs that were present in both cells, there were five other miRNAs detected as mutually dysregulated in MCF7-luminal and MDA-basal cancer cells. The expression of oncomiR miR-10b-5p and miR-486-5p was downregulated, and tumor suppressors such as miR-941, miR-200a-5p and miR-629-5p were upregulated accordingly. High expression of miR-629-5p has been linked to drug resistance and inhibition of metastasis and cancer growth, but none was related to the EMT regulation [38]. Meanwhile, the expression of miR-486-5p, which is cancer-type-specific, acts as a tumor suppressor in lung cancer and oncomiR in colorectal cancer [39]. It was found in this study that miR-486-5p expression was downregulated in both subtypes BCCs and can be classified as oncomiR as miR-486-5p were predicted to target tumor suppressor genes such as MAP3K7 and NFAT5. Recently, it was reported that miR-486-5p inhibited EMT by down-regulating SMAD2 and EMT regulators, vimentin and E-cadherin in breast cancer [40]. Reduced E-cadherin expression was attributed to increased miR-10b expression leading to the mesenchymal acquisition and vice versa [41]. As discussed earlier, miR-200a-5p that belongs to the miR-200 family can suppress EMT. In addition, Yu et al. (2018) demonstrated that miR-200a promoted chemoresistance by inhibiting DNA damage-induced apoptosis in breast cancer [42]. In another study, overexpression of miR-941 was shown to cause substantial inhibition of metastasis in hepatoma and gastric cancers [16,17]. To date, the discovery of those five miRNAs, particularly in EMT and dormancy regulation of breast cancer cells are still limited and more research is needed for a complete cancer remission.

Interestingly, both subtypes shared 7 out of 10 of the most enriched KEGG pathways despite having only 5 mutually dysregulated miRNAs. This implies that although different clusters of miRNAs were uniquely expressed in the co-culture of MCF7 and MDA, the enriched gene sets led to similarly associated pathways that were related to pathways of MET and tumor dormancy. The group of dysregulated miRNAs controlled cell dormancy by inhibiting mitogen-activated protein kinase (MAPK), TGF-β and Wnt signaling pathways, followed by the initiation of MET transcriptional regulation, epithelial phenotype stabilization and cellular proliferation and metastasis repression, which may lead to chemoresistance and relapse in treatment [26,43,44]. Other signaling pathways that can indirectly contribute to the aforementioned effects include ErbB, a pathway in cancer and neurotrophin signaling, which play a crucial role in the initiation and progression of many cancers [45,46]. Axon guidance, focal adhesion and insulin signaling pathways were reported to affect multiple cell types connected through the tumor microenvironment and act as the major mediator of signal transduction in the metastasis and promotion of tumors [47,48,49]. Subsequent activation of Toll-like-receptors (TLRs) in cancer cells and the resulting signaling cascade effect along with chemokine production could play an important role in promoting cancer cell survival and chemoresistance [50]. In addition, the participation of B and T cell receptors in the regulation of immune cells-mediated metastatic dormancy with the direct involvement of the immune system in protecting cancer cells was reported [51].

The present study identified signatures of breast cancer dormancy markers from the interaction of primary human adipose MSCs and breast cancer cell lines. The limitations of the use of cell lines in breast cancer research are well-documented, including the fact that immortalized cells may acquire genetic variances during the unlimited passaging compared to patient-derived tissues [52]. However, the use of cell lines is important as an in vitro model as it emerges as a feasible alternative to overcome the issues of sample size limitation and clearance consent from patients. In addition, the collection of tumor tissue of adequate quality for analysis is difficult to source, particularly the metastatic breast cancer tissue, as patients who develop the symptoms are normally already at stage IV. Moreover, establishing the procedures and facility for tissue collection is challenging to implement; still, it is required in order to ensure the collection of sufficient high-quality samples as assurance for the success of downstream molecular analysis [15]. Following advancements in cancer diagnostic using approaches such as transcriptome sequencing analysis of biomarkers, breast cancer can be detected before pathological symptoms develop. The use of more than single breast cancer-related markers can enhance the sensitivity and specificity of cancer detection for early diagnosis. This information can greatly contribute to the personalized clinical approaches in which specific treatment regimens can be designed based on the clinical history and stage of cancer as identified from profiles of markers discovered in cell lines [53]. Furthermore, the involvement of exosomes as biomarker carriers has greatly contributed to the advancement of breast cancer diagnostic and management [12,23,29]. Transition into the clinical phase requires the isolation of circulating exosomes from bodily fluids and serum, and their miRNA content has been shown to reflect their breast cancer cell origins [22]. It is therefore reasonable to rationalize the use of cell lines for assessing the progression of breast cancer and crucially contributing to clinical predictive values.

Among the mutually dysregulated miRNAs that are critically involved in the associated pathways of MET and dormancy, miR-941 was selected for further investigation. In the current study, the downstream effects of miR-941 overexpression in MCF7 and MDA cells were proven based on three pathways: inhibition of proliferation; inhibition of migration, invasion and wound healing; and promotion of MET. In agreement with findings from this study, miR-941 was shown to act as a tumor suppressor by suppressing cancer growth and migration of gastric and hepatoma cells via direct target of EMT inducers SNAIL1, ZEB1 and ZEB2 and promotes E-cadherin expression in the cancer cells [16,17]. It was confirmed that the expression of mesenchymal regulators was suppressed, and epithelial regulators were augmented using the qPCR technique. Association of miR-941 with cell proliferation and other biological processes through regulation of components involved in dormancy such as MAPK, TGF-β and Wnt signaling pathways was identified through bioinformatics analysis. However, the direct relationship between miR-941 and the dormancy was not confirmed. From the findings, exosomal transfer of miR-941 and its suppression of mesenchymal characteristic was proposed as one of the principal mechanisms contributing to the inhibition of cell proliferation and metastasis of luminal and basal BCCs subtypes as well as the induction of dormancy leading to cancer relapse.

Inhibition of breast cancer cell progression resulted from intercellular communication between adipose-derived stem cells and breast cancer cells have been reported many times, describing the changes in BCCs metastatic properties and phenotypes [13,54]. Despite that, this is the first study highlighting the miRNAs-dormancy markers that are unique and shared by both BC subtypes, MCF7 and MDA-MB-231. Although the question of when the dormant population of breast cancer transforms into an aggressive phenotype or stay permanently quiescent remains uncertain, we could show qualitative changes of BCCs in co-cultures with adipose MSCs associated with a more dormant-epithelial phenotype and lower rates of metastasis contributing to higher chemoresistance. Our findings alert the use of fat grafts supplemented with adipose MSCs for reconstruction or transplantation in patients after completed cancer treatment with regular screening to curb recurrence. For future study, the results need to be transferred to in vivo models to enlighten the interactions of breast cancer cells and adipose MSCs in a systemic environment.

## 4. Materials and Methods

### 4.1. Human Breast Cancer Cells Lines

The human breast cancer cell lines (BCCs), MCF7 (ATCC^®^ HTB-22™, passage number 10–30) and MDA-MB-231 (ATCC^®^ HTB-26™, passage number 10–30) were purchased from ATCC (Manassas, VA, USA) and maintained in RPMI-1640 supplemented with 10% exosome-depleted fetal bovine serum (System Biosciences, Palo Alto, CA, USA), 100 U/mL penicillin and 100 ug/mL streptomycin at 5% CO_2_ at 37 °C.

### 4.2. Adipose Mesenchymal Stem Cells

Primary human adipose MSC cells were obtained from Cryo Cord Sdn Bhd, and the research was approved by the UTAR Scientific and Ethical Review Committee (U/SERC/14/2012) in compliance with the guidelines regarding the use of primary stem cell for research. MSCs were maintained in MesenPRO RS medium (ThermoFisher Scientific, Arendalsvägen, Göteborg, Sweden) supplemented with 2 mM GlutaMAX (Invitrogen) at 5% CO_2_ at 37 °C. All cell lines were routinely mycoplasma tested using MycoAlertTM mycoplasma detection kit (Lonza, Morristown, NJ, USA). Phenotyping of adipose MSCs was performed by detecting the cell surface marker expression using fluorochrome-conjugated primary cocktail containing CD90+, CD44+ and CD105+ from FlowCellect human MSC characterization kit (Merck, New York, NY, USA). Following 30 min incubation of cells and antibody working solution in the dark at 4 °C, stained cells were acquired using a BD FACS CantoTM II flow cytometer (BD Biosciences, San Jose, CA, USA). The antibody panel used was composed of antibodies against CD90 CD44 and CD105 as positive MSC markers present on the surface of MSCs. The purity of MSC was determined as >85% positive for CD 90, CD 44 and CD105 surface antigen expression and later differentiated into adipocytes and osteoblasts [55]. The differentiated cells were stained with Oil Red O and Alizarin Red and recorded using an inverted microscope (Nikon, Melville, NY, USA).

### 4.3. Transwell Indirect Co-Culture Assay

Adipose MSCs were seeded in Transwell insert (Corning, Corning, NY, USA) containing polycarbonate membranes (1.0 μm pore size) at a density of 2 × 10^5^ cells/mL hanging over MCF7 or MDA-MB-231 cells that were plated in 6 well-plates separately at a density of 3 × 10^5^ cells/mL. The cells were co-cultured for 48 h. BCCs that were cultured alone were used as a negative control.

### 4.4. Internalization of Exosome and RNAs

MSCs were pre-stained with 5 µM of Vybrant-DiO or 1 µM of SYTO64 (Thermo Fisher Sc., UK) in a flask and incubated at 37 °C for 2 h. After removal of fluorescence dyes, stained cells were harvested and co-cultured with BCCs for 48 h. The uptake of exosomes and nucleic acids was observed using fluorescence microscopes (Carl Zeiss, Jena, Germany) with a green (484–501 nm) and a red wavelength (599–619 nm) filters. To track the direction of exosome and nucleic acid uptake, BCCs were pre-stained with Vybrant-DiO or SYTO-64, harvested, and co-culture with unstained MSC similarly (the opposite direction) [56].

### 4.5. Cell Proliferation Assay

Cell proliferation was detected using Cell counting kit-8 (CCK-8) (Sigma-Aldrich, St. Louis, MO, USA) according to the manufacturer’s instructions. A 100 µL of cells (1.0 × 10^4^ cells/well) was seeded in a 96-well plate and incubated for 24 h, 37⁰C, 5% CO2. After treatment, 10 µL of CCK-8 reagent was added to the cells and further incubated for 4 h. The results were measured using Infinite^®^ 200 PRO microplate reader (Tecan, Männedorf, Switzerland) at a wavelength of 450 nm.

### 4.6. Cell Cycle Analysis

After 48 h of co-culture, 1.0 × 10^6^ cells were harvested and stained with Cycletest™ Plus DNA kit (BD Biosciences, USA) containing reagent such as trypsin, trypsin inhibitor, RNAse and propidium iodide for 10 min each. Cell cycle distribution was run using BD FACS CantoTM II flow cytometer (Becton Dickinson, Franklin Lakes, NJ, USA) and further analyzed using BD CellQuest software (BD Biosciences, USA).

### 4.7. Transwell Invasion and Migration Assays

Invasion assay was carried out by coating 8.0 μm pore polycarbonate Transwell membrane inserts (Corning, USA) with Matrigel basement membrane matrix (BD Biosciences) [6]. Then, BCCs were seeded in a serum-free medium in the upper chamber, while the lower chamber contained a growth medium with 10% FBS or NIH-3T3 conditioned medium. NIH 3T3 fibroblasts is a sterile-filtered conditioned culture medium to stimulate the invasion and migration of MCF7 (low metastatic) cells. Within 48 h, cells were allowed to invade the Matrigel prior to fixing, permeabilization and staining with formaldehyde, methanol and crystal violet, respectively. Noninvaded cells at the surface of the membrane were scrapped-off using cotton swabs. Migration assay was performed similarly, without Matrigel. Images of cells that migrated and invaded through the membrane were captured using inverted microscopes (Nikon, USA).

### 4.8. Wound Healing Assay

Control and co-culture cells were seeded in 24-well plates at a density of 1.0 × 10^5^ cells/well and allowed to grow nearly confluent. Thereafter, a wound/gap was created in the center of the well, and the detached cells were rinsed with PBS. Wound/gap was monitored at 24 h interval (0 h, 6 h and 24 h), and images were taken using inverted microscopes in at least 5 fields (Nikon, USA). Percentage of migration was assessed by calculating the gap distance using Image J software (NIH) [57].

### 4.9. Chemoresistant MTT Assay

Cells were seeded into 96-well plates at a cell density of 5 × 10^3^ cells/100 μL 24 h. The cells were separately treated with doxorubicin, tamoxifen, cisplatin and 5-Hydroxy-[1,4]-Naphthoquinone (5HNQ), also known as juglone. After 48 h of incubation, 20 μL of MTT solution (5 mg/mL) was added into each well and further incubated for 4 h. The formed formazan precipitate was dissolved in DMSO before absorbance at 570 nm was taken with Infinite^®^ 200 PRO microplate reader. All experiments were carried out in triplicate. The IC50 was generated from the dose–response curves at 48 h.

### 4.10. 3D Spheres Formation Assay

Dissociated single-cell of control and co-culture BCCs were plated in a non-adherent ultra-low attachment plate, serum-free medium of DMEM/F12 supplemented with 20 ng/mL human recombinant epidermal growth factor and basic fibroblast growth factor (Invitrogen, USA), 1 × B27 (Thermo Scientific, Waltham, MA, USA), 10 µg/mL insulin, 0.5 µg/mL hydrocortisone, 100 U/mL penicillin and 100 µg/mL streptomycin and 0.4% bovine serum albumin (Sigma, Gillingham, Dorset, UK). Tumorspheres were maintained at 5% CO_2_, 37 °C. After 21 days, spheres with a diameter size larger than 50 µm were counted under an inverted microscope.

### 4.11. Flow Cytometric Analysis of CD24 and CD44 Surface Markers

Cells were harvested and resuspended in 100 μL buffers. Combinations of fluorochrome-conjugated monoclonal antibodies obtained from BD Biosciences (San Diego, CA, USA) against human CD44-FITC, CD24-PE and their respective isotype controls were added to the cell suspension as recommended by the manufacturer. After incubation at 4 °C in the dark for 30 min, cells were washed in the wash buffer and analyzed on a BD FACS CantoTM II flow cytometer (Becton Dickinson, USA).

### 4.12. Isolation of Exosomes

After 48 h of co-culture, the conditioned media was harvested through serial centrifugation at 300 g at 4 °C for 10 min and followed by 3000 g at 4 °C for 15 min to remove cell debris. The cell-free supernatant was pre-filtered using a 0.4 µm syringe followed by exosome isolation using exoEasy Maxi (Qiagen, Hilden, Germany) accordingly.

### 4.13. Exosomes Characterization

The acetylcholine esterase (AChE) enzyme activity that is commonly enriched within exosomes was detected using AChE colorimetric assay kit (Abcam, Cambridge, MA, USA). The exosomes were also observed for their morphology and protein surface markers (CD 63, CD 81 and CD 9) with immunogold-labeling using transmission electron microscopy (TEM), 80–120kVolt (Hitachi H7700, Hitachinaka, Ibaraki, Japan) [2].

### 4.14. Exosomes Functionality Study

To further confirm the role of exosomes mediating cross-talk between MSC and cancer cells in the microenvironment, exosomes from MSC and co-culture medium were collected. Aliquots of isolated exosomes were added into the medium culture of breast cancer cells to allow the uptake by cancer cells [6]. After 48 h of interaction, exosome functionality was screened on BCCs via proliferation, migration, invasion and wound healing assays, as well as qPCR of EMT-MET genes.

### 4.15. MicroRNA Expression Study

#### 4.15.1. Isolation of Cellular and Exosomal Total RNAs

For isolation of total RNAs, including small RNAs (miRNA), cells and exosomes were extracted using miRNeasy mini kit (Qiagen, Germantown, MD, USA). The quality and quantity of RNAs were measured using Agilent 2100 Bioanalyzer (Agilent, Santa Clara, CA, USA) and Qubit 2.0 Fluorometer (Invitrogen, Waltham, MA, USA), respectively.

#### 4.15.2. Construction of Small RNA Libraries and Sequencing

For small RNA-Sequencing using MiSeq (Illumina, San Diego, CA, USA), 100 ng of total RNA per sample was used for library construction using NEBNext^®^ Multiplex small RNA library prep set for Illumina (New England Biolabs, Ipswich, MA, USA) following manufacturer’s protocol. The resulting PCR products were purified using QIAquick PCR purification kit (Qiagen, Germany) and separated on a 6% Polyacrylamide gel, with miRNA size selection at the 140–150 bp range. The library was extracted from the gel using a DNA gel elution buffer. A total pool of 4 nM purified library with correct size was sequenced using MiSeq Reagent Kit v3 150 cycle and run using MiSeq System (Illumina, USA).

#### 4.15.3. Analysis of miRNA Sequencing Data

Raw reads were preprocessed by trimming the 3′ adapter using Cutadapt version 1.16 (Martin, 2011) and filtered using FASTX toolkit version 0.0.14 [58]. The remaining reads were inspected for quality control using FastQC software (http://www.bioinformatics.babraham.ac.uk/projects/fastqc/). Clean data were obtained by filtering out low-quality reads and bases with less than 18 and more than 26 nucleotides prior to mapping miRNA read sequences to the human genome. Reads were aligned to the human reference genome (UCSC 38) using Mapper and precursor mature miRNA reference (miRBase 21) using miRDeep2 version 2.0.0.8. [59]. The sequencing data are available in the NCBI Gene Expression Omnibus (GEO) database under the series accession identifier GSE156380.

#### 4.15.4. Prediction of Highly Dysregulated miRNAs Target Genes and Pathway Enrichment Analysis

A target gene prediction platform, miRSystem, was used to predict the highly dysregulated miRNAs target genes. The program integrates seven prediction platforms, namely DIANA, miRanda, miRBridge, PicTar, PITA, rna22, and TargetScan, allowing querying of multiple miRNAs for the associations between the miRNAs and their target genes and illustrating the enriched biological pathways, Kyoto Encyclopedia of Genes and Genomes (KEGG) Pathway [60]. To illustrate the gene ontology (GO) functional enrichment, the verified predicted target genes were performed using the Panther DNA database (http://pantherdb.org) [61].

#### 4.15.5. Verification of miRNA NGS Results by qRT-PCR

Based on sequencing data, 5 differentially expressed miRNAs (Table 1) were selected for validation using qRT-PCR. MicroRNAs were converted into cDNA using All-in-One™ miRNA First-Strand cDNA synthesis kit (GeneCopoeia, Rockville, MD, USA). Subsequently, the synthesized cDNAs were amplified using specific primers from GeneCopoeia (Table 3) and SYBR-Green mix (PCR Biosystems, UK) under the following condition; DNA Polymerase Activation at 95 °C for 2 min, followed by 40 cycles of 95 °C for 5 s and 62 °C for 25 s. Relative quantification was analyzed by geometric average Cq of the miRNAs normalized by the average of reference miRNAs (RNU6b). The differences in fold expression were calculated following the 2^−ΔΔCt^ method.

#### 4.15.6. Changes in the Expression of MDR, CSC, DNA Repair, Epithelial and Mesenchymal Genes

Total RNAs extracted were converted into cDNAs using iScript™ reverse transcription Supermix (BioRad, Hercules, CA, USA). Subsequently, the synthesized cDNAs were further diluted (200 ng) and amplified using the respective primers (Table 4, Integrated DNA Technologies, Singapore). The mRNA expression level was quantified using SYBR-Green mix (PCR Biosystems, London, UK) under the following condition; DNA polymerase activation at 95 °C for 2 min, followed by 40 cycles of 95 °C for 5 s and 62 °C for 25 s or TaqMan™ gene expression assays (Applied Biosystems, Foster City, CA, USA) under the following condition; UNG activation at 50 °C for 2 min DNA Polymerase Activation at 95 °C for 20 sec, followed by 40 cycles of 95 °C for 1 s and 60 °C for 20 s. Relative quantification was analyzed by geometric average Cq of the mRNAs normalized by the average of reference mRNAs (ACTB). The differences in fold expression were calculated following the 2^−ΔΔCt^ method.

#### 4.15.7. Transfection of miRNA Mimic

Commercial miR-941 mimic, MIMAT0004984:5′CACCCGGCUGUGUGCACAUGUGC-3′, AllStars Cell Death siRNA (Cat.no: 1027298) and AllStars negative control siRNA (Cat.no: 1027280) (Qiagen, USA) were used for transfection. The transfection was accomplished using HiPerFect transfection reagent (Qiagen, USA) according to the manufacturer’s protocol. Transfected cells were further incubated under normal conditions for 48 h, and downstream assays were applied.

### 4.16. Statistical Analysis

The data are presented as mean ± SD. For comparisons between two groups, the Student’s t-test was used. Comparisons among more than three groups were determined by one-way ANOVA followed by Bonferroni’s post hoc testing. All data were analyzed for significance using GraphPad version 8.0.1 with *p*-value < 0.05 was considered significant.

## 5. Conclusions

The outcome of this study offers new perspectives on how MCF7 and MDA-MB-231 cells enter a dormant state before the recurrence of cancer cell growth. Disrupting the pathways that govern the maintenance of dormancy may cause the cells to exit from dormancy, enter a proliferative state, and recurrence takes place. Therefore, the potential therapeutic approaches to avoid recurrence are by maintaining BCCs in a dormant state or blocking preferred pathways to enter the cell cycle arrest. It can be done by detecting miRNAs that regulate dormancy and metastasis and inhibit their expression before cancer relapse. In conclusion, the findings revealed the role of the miR-200 family, miR-146a-5p and miR-941 as molecular mediators of MCF7-luminal, MDA-basal subtypes and both subtypes, respectively. Owing to the complexity of exosomes contents, other small RNA populations such as tRNAs, unknown small RNAs, proteins and DNA, which are not highlighted in this study, may also contribute to the ability of BCCs to induce MET dormancy and chemoresistance, which require further research. To date, this is the first comprehensive microRNA expression profiling study of MCF7-luminal versus MDA-basal BCCs interaction with adipose MSCs involving exosomes. This research provides a reference framework for future studies focusing on aberrations of microRNA expression in human breast cancer and will contribute to a better understanding of differentiation lineages and malignant transformation of different breast cancer subtypes.

## Figures and Tables

**Figure 1 pharmaceuticals-14-00008-f001:**
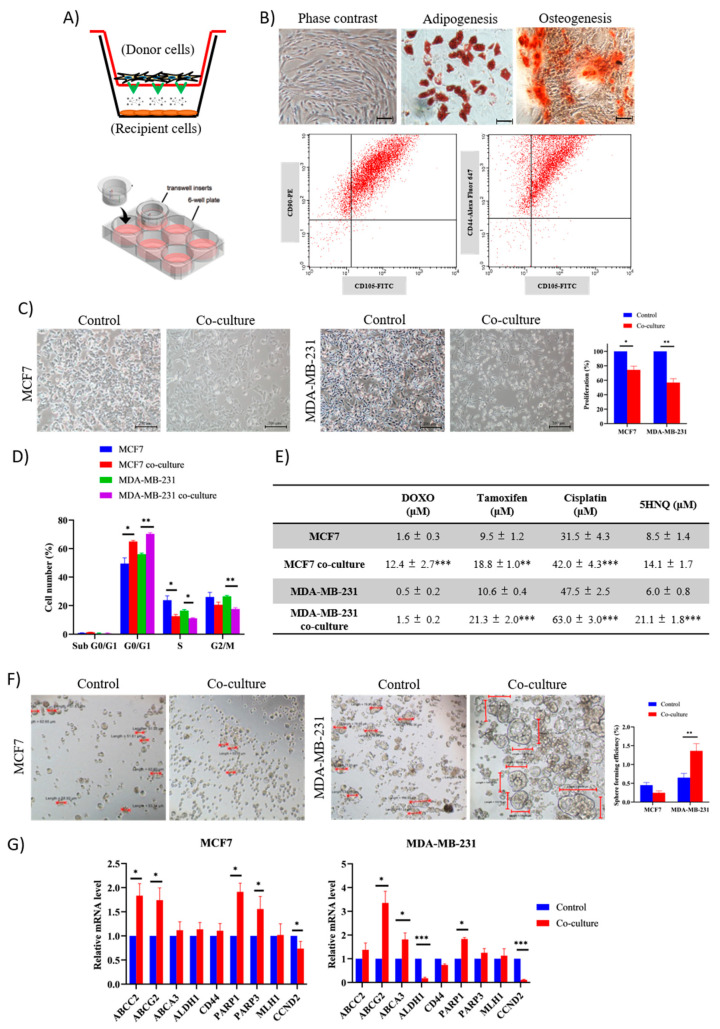
The inhibition effects of co-culture mesenchymal stem cell (MSC) on human breast cancer cell lines (BCCs) progression. (**A**) Schematic representation of Transwell indirect co-cultures between BCCs and adipose MSC containing polycarbonate membranes 1.0 μm pore size. (**B**) Characterization of adipose MSCs. Primary adipose MSCs after reaching confluence at passage 5. Adipogenic and osteogenic differentiation stained with Oil Red O dye to visualize the lipid droplets and Alizarin Red dye following differentiation. Scale bar = 100 μm. Representative graphs of adipose MSCs immunophenotyping were all positive for CD105, CD90 and CD44. (**C**) Inhibition of cancer cell proliferation (10× magnification). (**D**) Cell cycle G0/G1 phase arrest. (**E**) Decreased sensitivity to chemotherapy drugs (IC50 values of drugs in co-culture treated cells increased after 48 h of incubation). (**F**) Mammary tumors sphere formation in 3D culture (10× magnification). (**G**) Dysregulation of multiple drug resistance (MDR), cancer stem cell (CSC) and DNA repair genes expression. The results were analyzed using a t-test with mean values ± SD. * *p* < 0.05, ** *p* < 0.01 and *** *p* < 0.001 compared with the non-co-culture group.

**Figure 2 pharmaceuticals-14-00008-f002:**
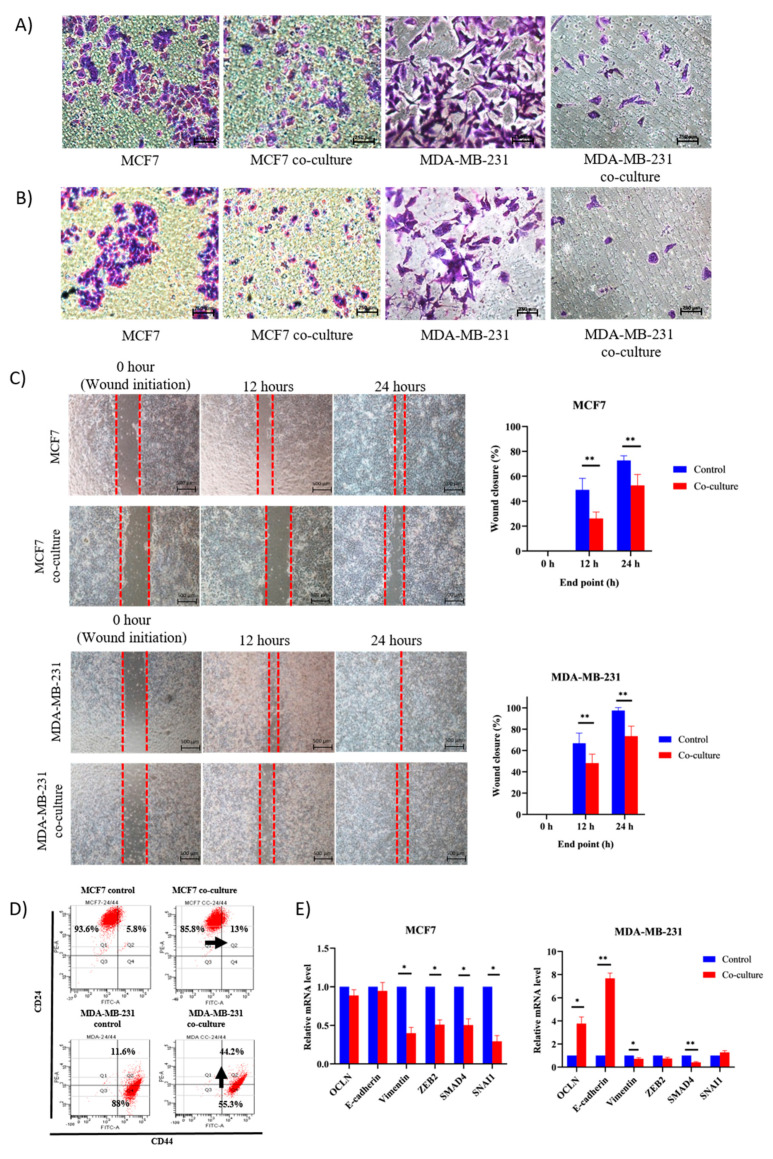
Metastasis capacity of cancer cells co-cultured with MSC. (**A**) Inhibition of migration assay (20× magnification). (**B**) Inhibition of invasion assay (20× magnification). (**C**) Decreased wound healing ability (10× magnification). The dotted line represents the corresponding edges approximations for the wound gap created using scratch technique. Ability of cells to migrate and close the gap represents their migration ability. (**D**) Alteration in epithelial (CD24) and mesenchymal (CD44) surface markers expressions. (**E**) Alteration in epithelial and mesenchymal genes expression level. The results were analyzed using a *t*-test with mean values ± SD. * *p* < 0.05 and ** *p* < 0.01 compared with the non-co-culture group.

**Figure 3 pharmaceuticals-14-00008-f003:**
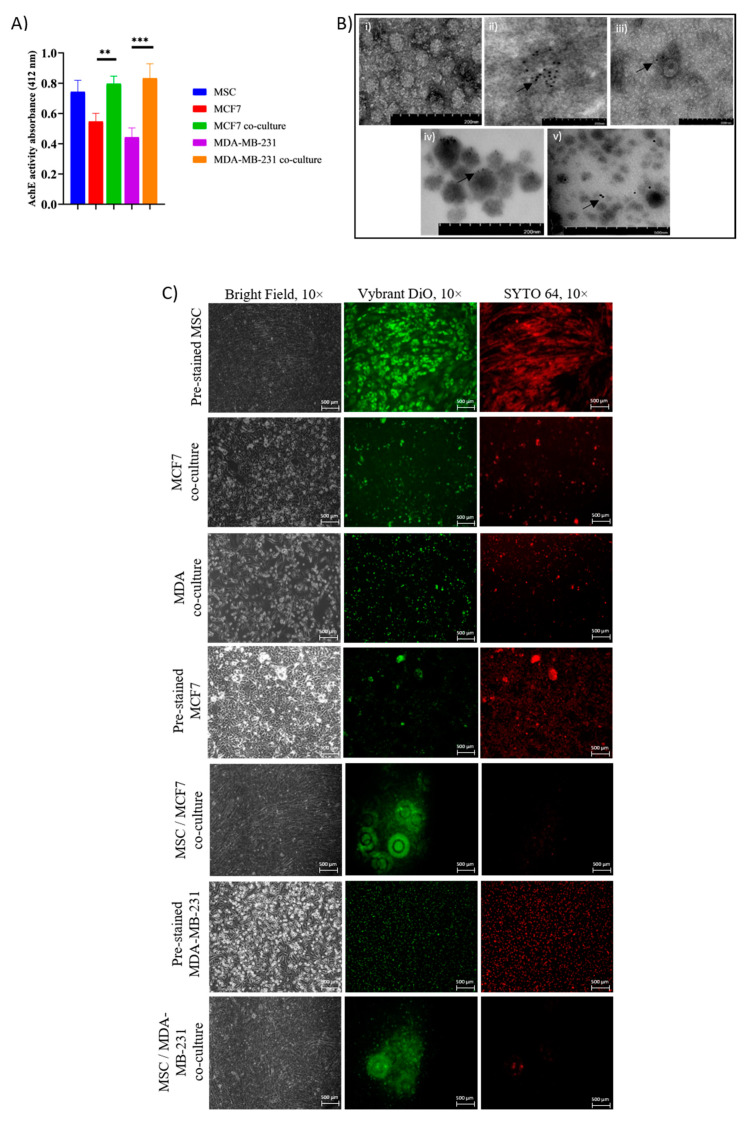
Characterization and intercellular transfer of exosomes. (**A**) A graph showing the enzymatic activity acetylcholine esterase (AchE) of isolated co-culture exosomes. (**B**) Representative image of transmission electron microscopy (TEM) of cells-derived exosomes; (i) negative staining-no primary antibody, (ii–v) exosomes stained with gold-conjugates secondary antibody to anti-CD81, anti-CD9, anti-CD63 and anti-TSG101 (80k× magnification). The arrow indicates the golden particles of the secondary antibody which represents the presence of protein markers CD81, CD9, CD63 and TSG101. (**C**) Intercellular transfer of exosomes and RNAs was visualized through membrane and RNAs staining with Vybrant DiO and SYTO 64 fluorescence dyes (10× magnification). The results were analyzed using ANOVA with mean values ± SD. ** *p* < 0.01 and *** *p* < 0.001 compared with the control group.

**Figure 4 pharmaceuticals-14-00008-f004:**
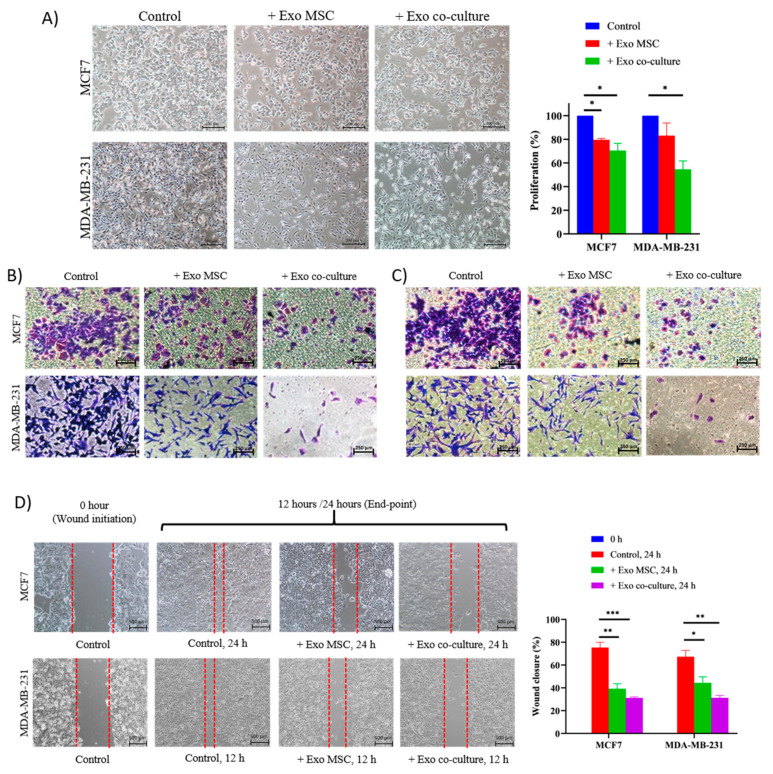
Analysis of BCC metastasis after interaction with exosomes derived from MSC and co-culture. The exosome functionality assays were conducted to observe the changes in (**A**) cell morphology and proliferation, (**B**) cell migration, (**C**) cell invasion, (**D**) wound healing. The results were analyzed using *t*-test and ANOVA with mean values ± SD. * *p* < 0.05, ** *p* < 0.01 and *** *p* < 0.001 compared with the control group.

**Figure 5 pharmaceuticals-14-00008-f005:**
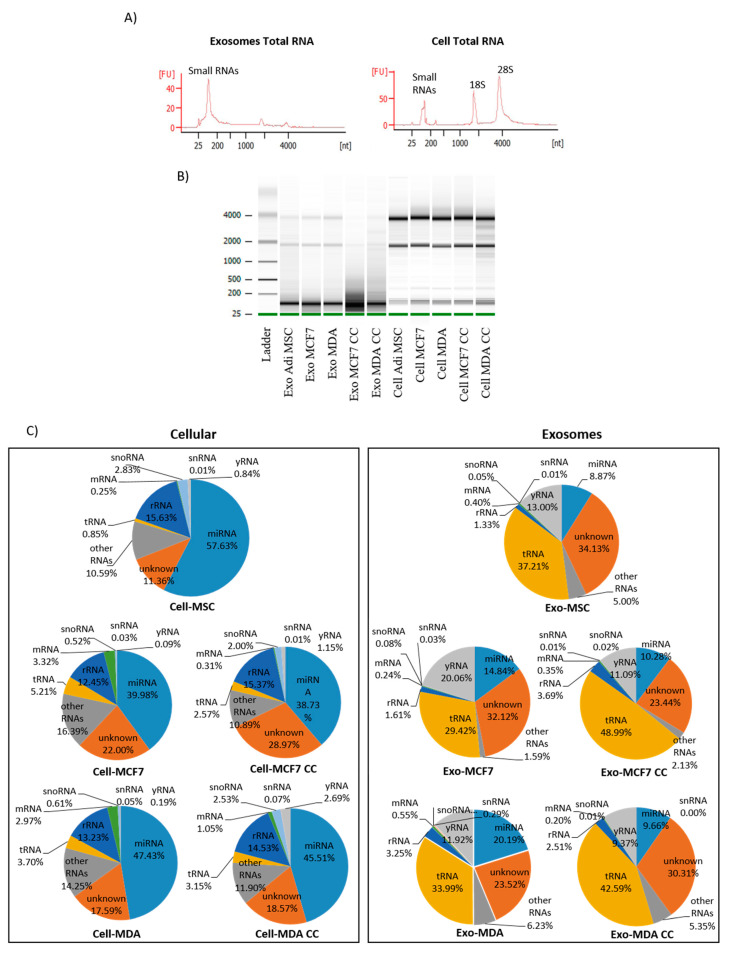
Characterization of RNAs and small RNAs composition (**A**) Representative peaks of total RNAs in cells and exosomes by bioanalyzer, respectively. Exosomes lack detectable 18S and 28S rRNAs compared to total cell RNAs; meanwhile, miRNAs population present in both at <40 nucleotides. (**B**) Length distribution of total RNAs in all samples of exosomes and cells. (**C**) Small RNAs annotation profiles of cells and exosomes predominantly consist of miRNA, tRNA, yRNA, mRNA and rRNA.

**Figure 7 pharmaceuticals-14-00008-f007:**
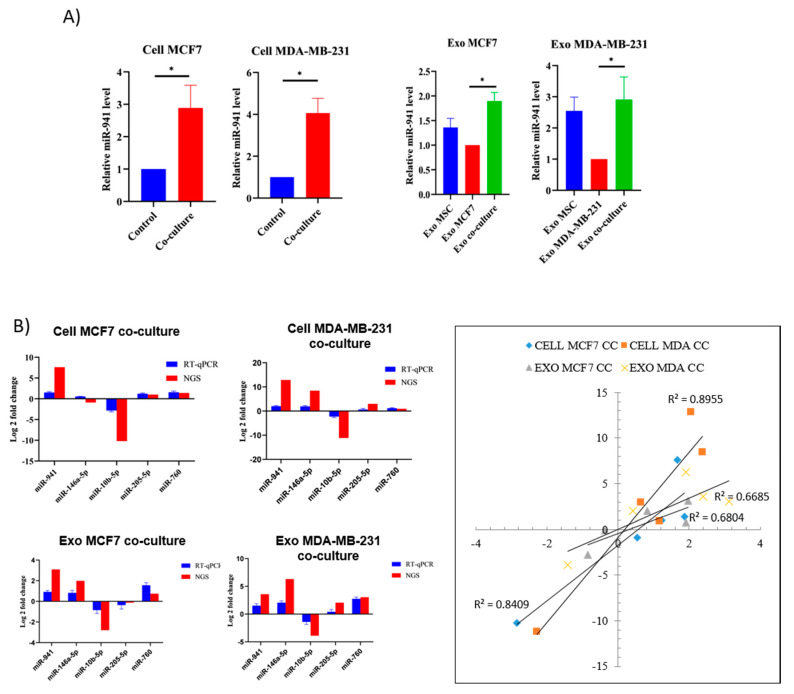
MiRNA-941 is specifically upregulated in co-culture BCCs and is a potential tumor suppressor miRNA diagnostic marker (**A**) Relative miR-941 expression in co-culture cells and exosomes (* *p* < 0.05) (**B**) RT-qPCR validation and correlation plot. Reproducibility of NGS and comparison of miRNA expression between NGS and qPCR analysis. The qPCR validation correlated well with sequencing data. Corresponding R2 values were determined by linear regression analysis.

**Figure 8 pharmaceuticals-14-00008-f008:**
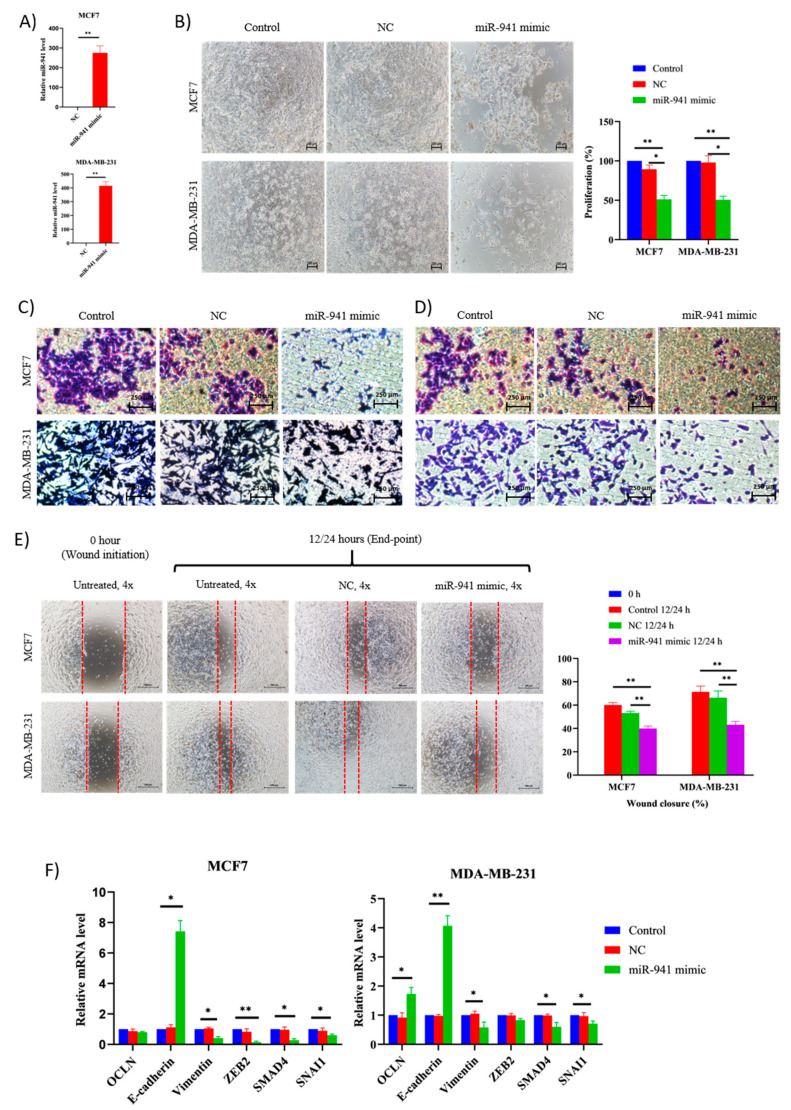
Functional analyses of miR-941 in co-culture subtypes MCF7 and MDA-MB-231 cells. (**A**) Augmentation of relative miR-941 expression in BCCs after 48 h of transient transfection. (**B**) Suppression of cell viability and proliferation of BCCs transfected with non-targeting negative control miRNA (AllStars negative control) or miR-941 mimic; (**C**) Suppression of migration assay; (**D**) Suppression of invasion assay; (**E**) Suppression of wound healing. Pictures were taken at 12 or 24 h later after wounding. The bar graph represents the percentage mean of wound closure of 5 random microscopic fields; (**F**) Alteration in epithelial and mesenchymal genes expression level. The results were analyzed using ANOVA with mean values ± SD. * *p* < 0.05, ** *p* < 0.01.

**Table 3 pharmaceuticals-14-00008-t003:** miRNA primers for RT-qPCR validation.

miRNA	Sequence Accession Number	Target Sequence
hsa-miR-146a-5p	MIMAT0000449	UGAGAACUGAAUUCCAUGGGUU
hsa-miR-941	MIMAT0004984	CACCCGGCUGUGUGCACAUGUGC
hsa-miR-10b-5p	MIMAT0000254	UACCCUGUAGAACCGAAUUUGUG
hsa-miR-760	MIMAT0004957	CGGCUCUGGGUCUGUGGGGA
hsa-miR-205-5p	MIMAT0000266	UCCUUCAUUCCACCGGAGUCUG

**Table 4 pharmaceuticals-14-00008-t004:** List of epithelial-to-mesenchymal transition (EMT)- mesenchymal-to-epithelial (MET) gene primers used in qRT-PCR.

Genes	Target Sequence (5′-3′)/ Assay ID
Housekeeping Reference Genes	
ACTB	FP: 5′-AGAGCTACGAGCTGCCTGAC-3′ RP: 5′-AGCACTGTGTTGGCGTACAG-3′
Epithelial genes	
E-cadherin	FP: 5′-ACAGGAACACAGGAGTCATCAG-3′ RP: 5′- CCCTTGTACGTGGTGGGATT-3′
OCLN	FP: 5′-TGCCTAGCTACCCCCATCTT-3′ RP: 5′-TGC ACC CAG CACAGATCAAT-3′
Mesenchymal Genes	
SNAIL	FP: 5′-AGTGGTTCTTCTGCGCTACTG-3′ RP: 5′-TGCTGGAAGGTAAACTCTGGATTAG-3′
SMAD4	FP: 5′-CTCCAGCTATCAGTCTGTCA-3′ RP: 5′-GATGCTCTGTCTTGGGTAATC-3′
Vimentin	FP: 5′-CCTGCAATCTTTCAGACAGG-3′ RP: 5′-CTCCTGGATTTCCTCTTCGT-3′
Zeb2	FP: 5′-TTTCAGGGAGAATTGCTTGA-3′ RP: 5′-CACATGCATACATGCCACTC-3′
Multidrug-resistant-ABC Transporter Genes	
ABCC2	FP: 5′-TGC AGC CTC CAT AAC CAT GAG-3′ RP: 5′-GAT GCC TGC CAT TGG ACC TA-3′
ABCG2	FP: 5′-CAG GTC TGT TGG TCA ATC TCA CA-3′ RP: 5′-TCC ATA TCG TGG AAT GCT GAA G-3′
ABCA3	FP: 5′-CAA AAC CCT GGA TCA CGT GTT-3′ RP: 5′-CCT CCG CGT CTC GTA GTT CT-3′
Cancer Stem Cell Genes	
ALDH1A1	FP: 5′-AGCAGGAGTGTTTACCAAAGA-3′ RP: 5′-CCCAGTTCTCTTCCATTTCCAG-3′
CD44	FP: 5′-CACAAATGGCTGGTACGTCTT-3′ RP: 5′-TTCATCTTCATTTTCTTCATTTGG-3′
DNA Repair Genes (TaqMan^®^ Probe)	
PARP1	Hs00242302_m1
PARP3	Hs00193946_m1
MLH1	Hs00979919_m1
CCND2	Hs00153380_m1

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
