# Peer review of "Adipose MSCs Suppress MCF7 and MDA-MB-231 Breast Cancer Metastasis and EMT Pathways Leading to Dormancy via Exosomal-miRNAs Following Co-Culture Interaction"

_pharmaceuticals, 2020, doi:10.3390/ph14010008_

Round 1
Reviewer 1 Report
Mohd Ali et al. present a very interesting work with a novel approach to research in breast cancer. The authors implement an adequate methodology with reliable results that support their conclusions. The discussion is proportional to the results shown and the current state of the art. For all these reasons, I consider the publication of this manuscript important. However, the authors should consider some important major and minor points in their manuscript:
-The summary should focus a little more on introducing breast cancer, since the journal does not have a word limit.
-The introduction has very relevant and appropriate information. However, the authors must restructure it. They should give more nuances at the beginning from the clinical point of view of breast cancer metastasis, as well as explicitly mention the differences in the incidence and epidmeiology of this complication. The miRNAs are entered in line 81, must be entered first. The authors should mention the current studies to focus the clinical translation that may have the spirit of their study.
-The material and methods must be expanded so that the results are reproducible. The authors must include in the commercial products, specific identifications of each substance and analyzer machines. In this sense, the authors should include more references in the phenotyping methods used. There are important points where information is lacking, such as the temperatures in Table 3. The authors must justify the statistical tests used, because there are other tests with greater force.
-The authors should consider that their results are shown in a very difficult to follow. Figures need to be divided, the information in some of them like figure 5 is not possible to read the information. Figures should be more self-explanatory, and since figures contain so much information, authors should improve and expand their figure legends. Histograms must be in color.
-The discussion is adequate. However, it should have a more clinical approach, the authors should argue with the large number of clinical trials that exist to date. The authors should discuss their limitations: as they should use a cell line of their own and make a correlation with clinical histories and survival of the patients. The authors should discuss the importance of the sample size and propose a possible pathological model. Authors should consult existing reviews to rework the discussion.
-The conclusion should be more explicit.
-The authors must check the English grammar. I suggest the authors to use the existing services.
Author Response
Response to Comments from Reviewer 1
Mohd Ali et al. present a very interesting work with a novel approach to research in breast cancer. The authors implement an adequate methodology with reliable results that support their conclusions. The discussion is proportional to the results shown and the current state of the art. For all these reasons, I consider the publication of this manuscript important. However, the authors should consider some important major and minor points in their manuscript:
Comment 1: The summary should focus a little more on introducing breast cancer, since the journal does not have a word limit.
Response: Introduction of breast cancer has been added into the summary/abstract section as highlighted below.
Abstract: Globally, breast cancer is the most frequently diagnosed cancer in women and it remains a substantial clinical challenge due to cancer relapse. The presence of a subpopulation of dormant breast cancer cells that survived chemotherapy and metastasized to distant organs may contribute to relapse. Tumor microenvironment (TME) plays a significant role as a niche in inducing cancer cells into dormancy as well as involves in the reversible epithelial-to-mesenchymal transition (EMT) into aggressive phenotype responsible for cancer-related mortality in patients. Mesenchymal stem cells (MSCs) are known to migrate to TME and interact with cancer cells via secretion of exosomes containing biomolecules, microRNA. Understanding of interaction between MSCs and cancer cells via exosomal miRNAs is important in determining the therapeutic role of MSC in treating breast cancer cells and relapse. In this study, exosomes were harvested from a medium of indirect co-culture of MCF7-luminal and MDA-MB-231-basal breast cancer cells (BCCs) subtypes with adipose MSCs. The interaction resulted in different exosomal miRNAs profiles that modulate essential signaling pathways and cell cycle arrest into dormancy via inhibition of metastasis and epithelial-to-mesenchymal transition (EMT). Overall, breast cancer cells displayed a change towards a more dormant-epithelial phenotype associated with lower rates of metastasis and higher chemoresistance. The study highlights the crucial roles of adipose MSCs in inducing dormancy and identified miRNAs-dormancy related markers. Thus, miRNAs secreted through exosomes from bidirectional interaction of MSCs and BCCs could be used to identify the metastatic pattern, predict relapses in cancer patients and to be potential candidate targets for new targeted therapy.
Comment 2: The introduction has very relevant and appropriate information. However, the authors must restructure it. They should give more nuances at the beginning from the clinical point of view of breast cancer metastasis, as well as explicitly mention the differences in the incidence and epidemiology of this complication. The miRNAs are entered in line 81, must be entered first. The authors should mention the current studies to focus the clinical translation that may have the spirit of their study.
Response: The information on Breast cancer metastasis incidence and the clinical view has been added and re-write in the introduction section as highlighted below. The abbreviation of miRNAs entered in line 81 has been introduced in the earlier paragraph in line 79.
[Line 54-59]
Metastatic or the spread of tumor cells throughout the body remains the underlying cause of death in the majority of breast cancer patients [2]. Patients present with distant metastasis are usually diagnosed with Stage IV disease and are unlikely treatable. Approximately 30% of women with breast cancer report recurrence with regional lymph node metastases despite early detection and advanced technology in cancer treatment due to the mechanism of resistance and tumor heterogeneity [3].
[Line 63-65]
The difference in the degree of tolerance to therapies among heterogeneous BCCs may render therapies in eliminating some subset of cancer cells such as dormant cells, contributing to late recurrence particularly in the high metastasis cells [5].
[Line 99-104]
From clinical practice, there are currently no available markers that are able to predict the risk of late recurrence and determine which dormant population will eventually develop aggressive phenotype or remain dormant [16]. Thus, there is a need for fundamental research in identifying molecular markers that are associated with the transition of cells in or out of the dormant stage for each breast cancer subtype to determine the prognosis and the therapeutic possibilities.
Comment 3: The material and methods must be expanded so that the results are reproducible. The authors must include in the commercial products, specific identifications of each substance and analyzer machines. In this sense, the authors should include more references in the phenotyping methods used. There are important points where information is lacking, such as the temperatures in Table 3. The authors must justify the statistical tests used, because there are other tests with greater force.
Response:
Some of the material and methods have already been expanded and highlighted in the manuscript.
The information on temperatures for Table 3 has been added in [line 810-812];
…under the following condition; DNA Polymerase Activation at 95°C for 2 min, followed by 40 cycles of 95 °C for 5 seconds and 62 °C for 25 seconds.
Also for Table 4 in [Line 821-826];
The mRNA expression level was quantified using SYBR-Green mix (PCR Biosystems, UK) under the following condition; DNA Polymerase Activation at 95°C for 2 min, followed by 40 cycles of 95 °C for 5 seconds and 62 °C for 25 seconds or TaqMan™ Gene Expression Assays (Applied Biosystems, USA) under the following condition; UNG activation at 50°C for 2 min DNA Polymerase Activation at 95°C for 20 sec, followed by 40 cycles of 95 °C for 1 second and 60 °C for 20 seconds.
Justification of statistical test used: For comparisons between two groups, the Student's t-test was used. The reason is that Student's' t-Test is one of the most commonly used techniques for testing a hypothesis on the basis of a difference between sample means. Next, comparisons among more than three groups were determined by one-way ANOVA followed by Bonferroni's post hoc testing. Bonferroni was used to reduce the instance of a false positive in which it designs an adjustment to prevent data from incorrectly appearing to be statistically significant.
Comment 4: The authors should consider that their results are shown in a very difficult to follow. Figures need to be divided, the information in some of them like figure 5 is not possible to read the information. Figures should be more self-explanatory, and since figures contain so much information, authors should improve and expand their figure legends. Histograms must be in color.
Response: Yes we agreed with the reviewer that some of the results shown in a complicated way. Figure 5 has been edited to a readable size and histograms have been changed to color. Figure legends also have been expended accordingly.
Comment 5: The discussion is adequate. However, it should have a more clinical approach, the authors should argue with the large number of clinical trials that exist to date. The authors should discuss their limitations: as they should use a cell line of their own and make a correlation with clinical histories and survival of the patients. The authors should discuss the importance of the sample size and propose a possible pathological model. Authors should consult existing reviews to rework the discussion.
Response: Discussion on the limitation of using cells line and not having a clinical approach has been discussed thoroughly in the manuscript highlighted below.
[Line 625-646];
The present study identified signatures of breast cancer dormancy markers from the interaction of primary human adipose MSCs and breast cancer cell lines. The limitations with the use of cell lines in breast cancer research are well-documented including the fact that immortalized cells may acquire genetic variances during the unlimited passaging compared to patient-derived tissues [52]. However, the use of cell lines is important as an in vitro model as it emerges as a feasible alternative to overcome the issues of sample size limitation and clearance consent from patients. In addition, the collection of tumor tissue of adequate quality for analysis is difficult to source particularly the metastatic breast cancer tissue as patients who develop the symptoms are normally already at stage IV. Besides, establishing the procedures and facility for tissue collection is challenging to implement, still, it is required in order to ensure the collection of sufficient high-quality samples as assurance for the success of downstream molecular analysis. Following advancements in cancer diagnostic using approaches such as transcriptome sequencing analysis of biomarkers, breast cancer can be detected before pathological symptoms develop. The use of more than single breast cancer-related markers can enhance the sensitivity and specificity of cancer detection for early diagnosis. This information can greatly contribute to the personalized clinical approaches in which specific treatment regimens can be designed based on the clinical history and stage of cancer as identified from profiles of markers discovered in cell lines [53]. Furthermore, the involvement of exosomes as biomarker carriers has greatly contributed to the advancement of breast cancer diagnostic and management [12,21,26]. Transition into the clinical phase requires the isolation of circulating exosomes from bodily fluids and serum, and their miRNA content has been shown to reflect the origin of breast cancer cells [27]. It is therefore reasonable to rationalize the use of cell lines for assessing the progression of breast cancer and crucially contributing to clinical predictive values.
Comment 6: The conclusion should be more explicit.
Response: The conclusion has been re-addressed as highlighted below.
[Line 663-674];
Inhibition of breast cancer cell progression resulted from intercellular communication between adipose-derived stem cells and breast cancer cells have been reported many times describing the changes in BCCs metastatic properties and phenotypes [13,54]. Despite that, this is the first study highlighting the miRNAs-dormancy markers that are unique and shared by both BC subtypes, MCF7 and MDA-MB-231. Although the question of when the dormant population of breast cancer transforms into an aggressive phenotype or stay permanently quiescent remains uncertain, we could show qualitative changes of BCCs in co-cultures with adipose MSCs associated with a more dormant-epithelial phenotype and lower rates of metastasis contributing to higher chemoresistance. Our findings alert the use of fat grafts supplemented with adipose MSCs for reconstruction or transplantation in patients after completed cancer treatment with regular screening to curb recurrence. For future study, the results need to be transferred to in vivo models to enlighten the interactions of breast cancer cells and adipose MSCs in a systemic environment.
Comment 7: The authors must check the English grammar. I suggest the authors to use the existing services.
Response: Thank you for your comments on the language. We have made some changes to the grammar mistakes yet, we would be happy to use the existing services as recommended.
Reviewer 2 Report
The authors present an interesting, well-structured manuscript with important results for understanding the behavior of this disease. Authors must make a series of changes so that their manuscript improves and some concerns that the reader may have must be resolved:
- The introduction is too long. Papers with a translational clinical focus should contain a connection between clinical practice and basic research in a more concise way. The authors must describe aspects related to the epidemiology of the disease, and how metastasis is related to these aspects. Afterwards, the authors should mention that studies are being carried out to identify the molecular aspects involved, introducing existing studies. This is how the authors justify their research.
- The results are described in a clear way, however the figures are complex to follow. I suggest that authors should make an effort to have a clearer connection so that the reader can interpret the authors' results. Figures 4 and 5 are intelligible.
- The discussion is well structured, the authors should take a more clinical approach, mentioning more specifically a shift between breast disease, cancer and metastasis. This connection is very necessary. Authors should include a discussion with other existing models. Authors must include data on human tissue.
- Authors should improve their materials and methods. Mention the typing of the cells. Information is missing in the RT-qPCR methodology.
- The authors could include human tissue, or at least mention this aspect among the limitations of the study.
- Authors should check their English grammar.
Author Response
Response to Comments from Reviewer 2
The authors present an interesting, well-structured manuscript with important results for understanding the behavior of this disease. Authors must make a series of changes so that their manuscript improves and some concerns that the reader may have must be resolved:
Comment 1: The introduction is too long. Papers with a translational clinical focus should contain a connection between clinical practice and basic research in a more concise way. The authors must describe aspects related to the epidemiology of the disease, and how metastasis is related to these aspects. Afterwards, the authors should mention that studies are being carried out to identify the molecular aspects involved, introducing existing studies. This is how the authors justify their research.
Response: The introduction has been re-write focusing more on clinical metastasis.
[Line 54-59]
Metastatic or the spread of tumor cells throughout the body remains the underlying cause of death in the majority of breast cancer patients [2]. Patients present with distant metastasis are usually diagnosed with Stage IV disease and are unlikely treatable. Approximately 30% of women with breast cancer report recurrence with regional lymph node metastases despite early detection and advanced technology in cancer treatment due to the mechanism of resistance and tumor heterogeneity [3].
[Line 63-65]
The difference in the degree of tolerance to therapies among heterogeneous BCCs may render therapies in eliminating some subset of cancer cells such as dormant cells, contributing to late recurrence particularly in the high metastasis cells [5].
[Line 99-104]
From clinical practice, there are currently no available markers that able to predict the risk of late recurrence and determine which dormant population will eventually develop aggressive phenotype or remain dormant [16]. Thus, there is a need for fundamental research in identifying molecular markers that are associated with the transition of cells in or out of the dormant stage for each breast cancer subtype to determine the prognosis and the therapeutic possibilities.
Comment 2: The results are described in a clear way, however, the figures are complex to follow. I suggest that authors should make an effort to have a clearer connection so that the reader can interpret the authors' results. Figures 4 and 5 are intelligible.
Response: Yes we agreed with the reviewer that some of the figures shown in a complicated way. Figures 4 and 5 have been edited to a readable size and arranged in a clearer way.
Comment 3: The discussion is well structured, the authors should take a more clinical approach, mentioning more specifically a shift between breast disease, cancer and metastasis. This connection is very necessary. Authors should include a discussion with other existing models. Authors must include data on human tissue.
Response: Discussion on the connection between breast disease, cancer and metastasis have been added accordingly.
[Line 433-439];
Breast cancer cells are likely to disseminate during cancer progression and interact with MSCs in TME releasing molecular signals contributing to the emergence of a subpopulation of cells that may go into the dormancy stage. The subpopulation of cell is able to avoid chemotherapy treatment and reawaken in response to signals and transform into the aggressive phenotype, resulting in recurrence and metastasis. The risk of developing distant metastasis may vary over time across different subgroups of breast cancer cells.
Discussion with other existing models also been added.
[Line 440-454];
Several types of in-vitro co-culture models have been developed to study the interaction between cancer cells and microenvironment components such as extracellular matrix (ECM), fibroblast and MSCs where these components have demonstrated the importance of the microenvironment in governing cancer cell growth [17-19]. The most common interaction models used to elucidate the interaction between two distinct cells includes direct co-culture [17] and indirect co-culture using culture insert separating cells in space but sharing similar culture condition, similar with our study [20]. There are also previous studies that have been carried out using conditioned mediums from MSCs and were incubated with cancer cells [21,22]. However, this method lacks the feedback-loop dependent signaling or bi-directional cross-talk. There is also a 3D co-culture model established to recapitulate the complexity of interaction between BCCs and bone marrow MSCs where it promotes dormancy [19,23]. Even though in vivo animal is a better model in representing the complexity and significance of the interaction between the cells, in vitro model is still required for preliminary study. Very few models have been developed concentrating on cellular dormancy and metastasis property [12,20]. Despite remarkable findings, the interaction effects of MSCs on BCC subtypes remain unanswered.
Data on human tissue is not available in our study, however, it has been addressed as a limitation in the discussion.
[Line 625-646];
The present study identified signatures of breast cancer dormancy markers from the interaction of primary human adipose MSCs and breast cancer cell lines. The limitations with the use of cell lines in breast cancer research are well-documented including the fact that immortalized cells may acquire genetic variances during the unlimited passaging compared to patient-derived tissues [52]. However, the use of cell lines is important as an in vitro model as it emerges as a feasible alternative to overcome the issues of sample size limitation and clearance consent from patients. In addition, the collection of tumor tissue of adequate quality for analysis is difficult to source particularly the metastatic breast cancer tissue as patients who develop the symptoms are normally already at stage IV. Besides, establishing the procedures and facility for tissue collection is challenging to implement, still, it is required in order to ensure the collection of sufficient high-quality samples as assurance for the success of downstream molecular analysis. Following advancements in cancer diagnostic using approaches such as transcriptome sequencing analysis of biomarkers, breast cancer can be detected before pathological symptoms develop. The use of more than single breast cancer-related markers can enhance the sensitivity and specificity of cancer detection for early diagnosis. This information can greatly contribute to the personalized clinical approaches in which specific treatment regimens can be designed based on the clinical history and stage of cancer as identified from profiles of markers discovered in cell lines [53]. Furthermore, the involvement of exosomes as biomarker carriers has greatly contributed to the advancement of breast cancer diagnostic and management [12,21,26]. Transition into the clinical phase requires the isolation of circulating exosomes from bodily fluids and serum, and their miRNA content has been shown to reflect their origin of breast cancer cells [27]. It is therefore reasonable to rationalize the use of cell lines for assessing the progression of breast cancer and crucially contributing to clinical predictive values.
Comment 4: Authors should improve their materials and methods. Mention the typing of the cells. Information is missing in the RT-qPCR methodology.
Response: In the manuscript, we have elaborated more on the immunophenotyping of human primary adipose MSCs using Fluorescence-activated cell sorter. The phenotyping was performed prior to co-culture for identification and characterization of the primary cells harvested from the adipose tissue donors.
[Line 687-694].
Phenotyping of adipose MSCs was performed by detecting the cell surface marker expression using fluorochrome-conjugated primary cocktail containing CD90+, CD44+ and CD105+ from Flowcellect human MSC characterization kit (Merck, USA). Following 30 minutes incubation of cells and antibody working solution in dark at 4°C, stained cells were acquired using a BD FACS CantoTM II Flow Cytometer (BD Biosciences, USA). The antibody panel used was composed of antibodies against CD90 CD44 and CD105 as positive MSC markers present on the surface of MSCs. The purity of MSC was determined as >85% positive for CD 90, CD 44 and CD105 surface antigen expression and later differentiated into adipocytes and osteoblasts [53].
As for breast cancer cell lines, both subtypes MCF7 and MDA-MB-231 were purchased from ATCC (VA, USA) and were maintained in low passage prior to use. The authentication of the cells has been verified by ATCC and is not an issue.
[Line 677-678];
MCF7 (ATCC® HTB-22™, passage number 10-30) and MDA-MB-231 (ATCC® HTB-26™, passage number 10-30)
As for RT-qPCR methodology, we have added temperature condition for Table3 [Line 810-812];
under the following conditions; DNA Polymerase Activation at 95°C for 2 min, followed by 40 cycles of 95 °C for 5 seconds and 62 °C for 25 seconds.
Comment 5: The authors could include human tissue, or at least mention this aspect among the limitations of the study.
Response: Thank you for the suggestion. We have included the limitation of using human tissue in the manuscript as highlighted below.
[Line 625-646];
The present study identified signatures of breast cancer dormancy markers from the interaction of primary human adipose MSCs and breast cancer cell lines. The limitations with the use of cell lines in breast cancer research are well-documented including the fact that immortalized cells may acquire genetic variances during the unlimited passaging compared to patient-derived tissues [52]. However, the use of cell lines is important as an in vitro model as it emerges as a feasible alternative to overcome the issues of sample size limitation and clearance consent from patients. In addition, the collection of tumor tissue of adequate quality for analysis is difficult to source particularly the metastatic breast cancer tissue as patients who develop the symptoms are normally already at stage IV. Besides, establishing the procedures and facility for tissue collection is challenging to implement, still, it is required in order to ensure the collection of sufficient high-quality samples as assurance for the success of downstream molecular analysis. Following advancements in cancer diagnostic using approaches such as transcriptome sequencing analysis of biomarkers, breast cancer can be detected before pathological symptoms develop. The use of more than single breast cancer-related markers can enhance the sensitivity and specificity of cancer detection for early diagnosis. This information can greatly contribute to the personalized clinical approaches in which specific treatment regimens can be designed based on the clinical history and stage of cancer as identified from profiles of markers discovered in cell lines [53]. Furthermore, the involvement of exosomes as biomarker carriers has greatly contributed to the advancement of breast cancer diagnostic and management [12,21,26]. Transition into the clinical phase requires the isolation of circulating exosomes from bodily fluids and serum, and their miRNA content has been shown to reflect their origin of breast cancer cells [27]. It is therefore reasonable to rationalize the use of cell lines for assessing the progression of breast cancer is very crucial as it contributes to clinical predictive values.
Comment 6: Authors should check their English grammar.
Response: Thank you for your comments on the language. We have made some changes to the grammar mistakes yet, we would be happy to use the existing services as recommended.